# MSTN: A Lightweight and Fast Model for General Time-Series Analysis

**Sumit S. Shevtekar**                                                        *sumit.shevtekar@gmail.com*
*Department of Computer Science and Engineering*
*Indian Institute of Technology Indore*
*Indore, 453552, Madhya Pradesh, India*

**Chandresh K. Maurya**                                                        *chandresh@iiti.ac.in*
*Department of Computer Science and Engineering*
*Indian Institute of Technology Indore*
*Indore, 453552, Madhya Pradesh, India*

**Reviewed on OpenReview:** *https://openreview.net/forum?id=je2N2nnDry*

## Abstract

Real-world time series often exhibit strong non-stationarity, complex nonlinear dynamics, and behavior expressed across multiple temporal scales, from rapid local fluctuations to slow-evolving long-range trends. However, many contemporary architectures impose rigid, fixed-scale structural priors—such as patch-based tokenization, predefined receptive fields, or frozen backbone encoders—which can over-regularize temporal dynamics and limit adaptability to abrupt high-magnitude events. To handle this, we introduce the *Multi-scale Temporal Network* (MSTN), a hybrid neural architecture grounded in an *Early Temporal Aggregation* principle. MSTN integrates three complementary components: (i) a multi-scale convolutional encoder that captures fine-grained local structure; (ii) a sequence modeling module that learns long-range dependencies through either recurrent or attention-based mechanisms; and (iii) a self-gated fusion stage incorporating squeeze–excitation and a single dense layer to dynamically reweight and fuse multi-scale representations. Importantly, MSTN applies early temporal aggregation immediately after encoding, ensuring that all subsequent refinement and prediction modules operate in constant time $\mathcal{O}(1)$ with respect to sequence length, while the front-end encoder retains its original complexity ($\mathcal{O}(L^2)$ for Transformer, $\mathcal{O}(L)$ for BiLSTM). This design enables MSTN to flexibly model temporal patterns spanning milliseconds to extended horizons, while avoiding the computational burden typically associated with long-context models. Across extensive benchmarks covering imputation, long-term forecasting, classification, and cross-dataset generalization, MSTN achieves state-of-the-art performance, establishing new best results on **21 of 27** datasets, while remaining lightweight ($\sim$0.40M params for MSTN-BiLSTM and $\sim$1.06M for MSTN-Transformer) and suitable for low-latency inference (**<1 sec, often in milliseconds**), resource-constrained deployment. Code: **https://github.com/SumitPTW/MSTN**

## 1 Introduction

Multivariate time series (MTS) analysis underpins a broad spectrum of societally and industrially critical applications, including healthcare monitoring (Morid et al., 2023), intelligent transportation systems (Kadiyala & Kumar, 2014), climate and environmental modeling (Gruca et al., 2022), energy systems (Kardakos et al., 2013), industrial prognostics, and behavioural analytics. Across these domains, learning from temporal data is central to tasks such as long-term forecasting, missing-value imputation, and trajectory or activity classification (Wu et al., 2022; Zhao et al., 2024). Despite decades of study, robust temporal modeling remains fundamentally challenging. Unlike language or vision, individual time points in a time series carry limited

semantic meaning; useful information emerges only through temporal variation patterns such as continuity, periodicity, and long-term trends (Wu et al., 2023; Lim & Zohren, 2021). This intrinsic dependency structure renders time series modeling both information-rich and computationally demanding.

Deep learning (DL) has substantially advanced time series analysis, not only improving predictive accuracy but also enabling transferable representations for downstream tasks (Nie et al., 2023). However, recent evidence that simple linear models can rival or even surpass sophisticated Transformer-based architectures on standard benchmarks (Zeng et al., 2022) has exposed structural limitations in contemporary designs. Classical neural architectures—including convolutional networks (Franceschi et al., 2019; Bai et al., 2018) and recurrent models with gating mechanisms (Hochreiter & Schmidhuber, 1997; Lai et al., 2018)—are constrained by vanishing gradients, limited parallelism, or restricted receptive fields. Although temporal convolutional networks improve efficiency (He & Zhao, 2019), they rely on fixed receptive scales. Transformer-based models promise global dependency modeling (Vaswani et al., 2023), yet their quadratic complexity in sequence length has driven a proliferation of sparsification and approximation strategies (Zhou et al., 2021; 2022). Many such methods continue to treat time steps as independent tokens, limiting their ability to encode local semantic structure and multi-scale temporal interactions.

These design pressures have led to the emergence of three dominant modeling paradigms, formalized as Channel Strategies (Qiu et al., 2025): Channel Independence (CI), Channel Dependence (CD), and Channel Partiality (CP). CI-based approaches prioritize computational efficiency by processing each variable independently, but sacrifice inter-variable interactions. CD-based models explicitly capture cross-channel dependencies at the cost of poor scalability. CP-based methods seek a compromise by projecting channels into lower-dimensional latent spaces, improving efficiency while retaining partial correlations. Despite their differences, all three paradigms rely on rigid structural assumptions—fixed patch sizes, predefined resolutions, uniform-scale mixing MLPs, static backbones, or explicit periodic decompositions—which limit adaptability to non-stationary, irregular, and long-sequence dynamics. As a result, even state-of-the-art (SOTA) systems such as PatchTST (Nie et al., 2023), TimesNet (Wu et al., 2023), iTransformer (Liu et al., 2024), TSMixer (Chen et al., 2023), and recent LLM-inspired approaches (Chang et al., 2025; Jin et al., 2024) struggle to maintain temporal fidelity under complex, real-world conditions.

A key limitation shared by these approaches is the implicit assumption of a single dominant temporal scale. This limitation is architecturally evident in designs such as TSMixer, where the time-mixing MLPs are shared across all time steps, resulting in a fixed-resolution, uniform transformation across the temporal dimension. Real-world temporal processes, however, are inherently multi-scale: micro-level fluctuations, sudden spikes, intermediate dynamics, and long-term trends often coexist and interact. Capturing such a hierarchical structure is essential in safety-critical and high-variability domains, including physiological monitoring, risky driving behavior analysis, and mechanical prognostics. Consequently, fixed-scale or uniform-scale architectures are ill-suited to these settings. Existing models handle a single task or focus on a single type of temporal variations- short or long, rarely balancing both effectively. There is a need for architectures that are not only general-purpose but also lightweight and efficient.

To address these challenges, we introduce the *Multi-scale Temporal Network* (MSTN), a hybrid architecture grounded in the principle of *Early Temporal Aggregation* (ETA). The central idea of ETA is to decouple expressive temporal modeling from inference-time complexity by collapsing the temporal dimension early in the network via an $L \rightarrow 1$ transformation, where $L$ denotes the lookback-window length. By performing the computationally intensive operations (e.g., $\mathcal{O}(L)$ BiLSTM or $\mathcal{O}(L^2)$ Transformer self-attention) *before* this aggregation step, the subsequent refinement layers—including feature fusion, SE recalibration, single dense layer (SDL), and prediction modules—operate with a fixed $\mathcal{O}(1)$ cost with respect to $L$. Importantly, while ETA makes the downstream modules constant in time $\mathcal{O}(1)$, the front-end encoder retains its original complexity ($\mathcal{O}(L^2)$ for Transformer, $\mathcal{O}(L)$ for BiLSTM). MSTN achieves this through a dual-path encoder that integrates complementary temporal strategies. Multi-scale convolutional encoders capture fine-grained local patterns, while a sequence modeling backbone (BiLSTM or Transformer) captures long-range dependencies. Their outputs are fused through a self-gated fusion (SGF) mechanism, augmented with squeeze-and-excitation (SE) recalibration and SDL, producing compact, discriminative, and temporally coherent representations.

We evaluate MSTN across forecasting, imputation, classification, and cross-domain generalization tasks, demonstrating consistent SOTA performance on 21 of 27 benchmark datasets. MSTN establishes new best results in long-term forecasting, achieves strong robustness under high missingness in imputation, and delivers improved classification accuracy across diverse application domains. In particular, these gains are achieved alongside efficiency improvements: the MSTN-BiLSTM and MSTN-Transformer variants utilize fixed cores of $\sim$398,008 parameters and $\sim$1,055,416 $\approx$ 1M parameters, respectively. The proposed work thus has a twofold impact. First, MSTN provides fast inference and can be embedded in real-time applications for time-series classification/forecasting, such as healthcare monitoring, industrial machine health, flight navigation, etc. Second, MSTN has a very small memory footprint and is thus suited for edge-device applications.

The key contributions of this work are summarized as follows:

1. We introduce MSTN, a multi-scale neural framework that is both fast and efficient. By leveraging the ETA principle to confine expensive $\mathcal{O}(L^2)$ operations to the initial encoder, MSTN ensures that all subsequent refinement and prediction modules operate in constant $\boldsymbol{O(1)}$ time with respect to sequence length, while the front-end encoder retains its original complexity ($\mathcal{O}(L^2)$ for Transformer, $\mathcal{O}(L)$ for BiLSTM). MSTN incurs $\sim$**398,008** (MSTN-BiLSTM) and $\sim$**1,055,416** (MSTN-Transformer) parameters and **sub-millisecond inference** latency.

2. Through an extensive evaluation across 27 standard benchmarks spanning imputation, long-term forecasting, classification, and cross-dataset generalization, MSTN achieves **SOTA performance on 21 datasets,** outperforming recent models such as GPT2, TimesNet, iTransformer, PatchTST, and TSMixer .

3. We also show the generalizability of MSTN on **7 datasets** from various domains such as health, safety, etc.

## 2 Related Work

In multivariate time series, a lot of work has been done. However, we are going to discuss only the most recent work in the literature of multivariate time series. Our discussion focuses on two aspects of time-series models: (i) the tasks they handle and (ii) efficiency.

### 2.1 Task Perspective

Work in time-series prediction can be categorized based on the following tasks: Imputation, long-term forecasting, classification, and anomaly detection. In this section, we focus on the first four tasks.

The majority of works focus on long-term forecasting tasks, including specialized architectures — such as PatchTST (Nie et al., 2023), iTransformer (Liu et al., 2024), TSMixer (Chen et al., 2023), TIME-LLM (Jin et al., 2024), LLM4TS (Chang et al., 2025)), and masked autoencoders (HiMTM (Zhao et al., 2024)). Seg-MoE (Ortigossa & Segal, 2026) improves scalability and long-term dependency modeling via segment-wise Mixture-of-Experts routing, but is primarily designed for forecasting and does not explicitly handle irregular or event-driven patterns. vLinear (Yue et al., 2026) introduces an efficient linear multivariate forecaster, achieving strong performance; however, its linear formulation limits its capacity to capture complex nonlinear temporal dynamics. These models have driven significant performance gains, but have not been applied to other time-series tasks like imputation or classification, which we address in the present work. In addition, the aforementioned works claim to handle channel dependency (CI, CD, or CP) differently. CI works like DLinear (Zeng et al., 2022) and PatchTST (Nie et al., 2023) handle time-series where channels are independent and therefore cannot model cross-channel effects. CD models such as iTransformer (Liu et al., 2024), graph-based approaches such as TPGNN (Liu et al., 2022b) capture full cross-channel dependencies but scale poorly with channel count. CP methods like MCformer (Han et al., 2024) and ModernTCN (donghao & wang xue, 2024) project the channel dimension into a latent space before interaction, trading expressivity for efficiency. While these methods show impressive performance, they are limited in capturing the multi-scale

temporal dynamics within and across channels—a capability essential for modeling long-sequence dependencies. To combine the strengths of both paradigms, our work takes an intermediate route where we leverage both CI and CD.

Another line of work in time-series prediction is imputation, which addresses the challenge of reconstructing missing values caused by sensor failures or data corruption, an essential step to maintain data integrity in real-world systems (Zhou et al., 2023; Wu et al., 2023). Imputation tasks employ specialized architectures such as TimesNet (Wu et al., 2023), GPT2(3) (Zhou et al., 2023), PatchTST (Nie et al., 2023), LightTS (Zhang et al., 2022), and DLinear (Zeng et al., 2022). However, these methods perform poorly on aperiodic or irregularly sampled events (e.g., emergency braking) because they have not been designed to detect such chaotic events. On the other hand, in the present work, MSTN can handle such events gracefully.

Another time series task, classification, involves categorizing the entire sequence to identify underlying events or patterns, enabling automated diagnosis and monitoring in fields such as healthcare and industrial maintenance (Wu et al., 2023; Zhang et al., 2022). Classification specialized architectures such as GPT2(6) (Alharthi et al., 2025), TimesNet (Wu et al., 2023), DLinear (Zeng et al., 2022), ETSFormer (Woo et al., 2022), FEDformer (Zhou et al., 2022), and Informer (Zhou et al., 2021). However, by design, these methods focus on seasonal decomposition, which can lead to over smoothing of temporal dynamics. Consequently, they often miss sudden, aperiodic anomalies such as spikes or crashes. They also tend to specialize in either local or global trends, rarely balancing both effectively.

## 2.2 Efficiency Perspective

In this section, we discuss efficiency of the time series models in terms of (i) characteristics of time series captured such as periodicity, varying temporal patterns, non-stationary behaviors, etc.; (ii) inference latency: and (iii) memory.

Periodic models such as TimesNet (Wu et al., 2023), Autoformer (Wu et al., 2022), ETSFormer (Woo et al., 2022), FEDformer (Zhou et al., 2022), LightTS (Zhang et al., 2022), PatchTST (Nie et al., 2023), and DLinear (Zeng et al., 2022) explicitly model periodic or seasonal structures and have demonstrated strong performance on long-term forecasting tasks. However, they often struggle to capture aperiodic, irregular, sudden spikes or event-driven dynamics. Dilated convolutional networks (e.g., ModernTCN (donghao & wang xue, 2024)) extract multiscale local patterns but have limited receptive fields for very long dependencies. Graph-based methods such as GTS (Shang et al., 2021) and FourierGNN (Yi et al., 2023) model spatio-temporal relationships but often assume static graphs, struggling with dynamically changing sensor relationships. Efficient attention variants (Informer (Zhou et al., 2021), FEDformer (Zhou et al., 2022), Autoformer (Wu et al., 2022)) use sparse or frequency-domain attention for long-range modeling. Despite these advances, these methods maintain the full sequence length throughout the network, incurring substantial memory and computational cost as the length of the sequence increases. They also tend to specialize in either local or global trends, rarely balancing both effectively while meeting the requirement for low-latency inference.

Recent MLP-based architectures such as TSMixer (Chen et al., 2023) and TTM (Ekambaram et al., 2024) achieve computational efficiency through temporal and channel mixing, demonstrating competitive performance with lower computational costs compared to attention-based models. However, these architectures employ fixed mixing operations that may not adapt dynamically to varying temporal patterns across different scales and frequencies. A distinct line of work employs large language models (LLM) for time series (e.g., TIME-LLM (Jin et al., 2024), LLM4TS (Chang et al., 2025)), leveraging their few-shot capabilities for cold-start scenarios. However, these models suffer from high latency and memory footprints, making them unsuitable for real-time edge deployment. Recent architectures such as iTransformer (Liu et al., 2024) improve multivariate dependency modeling by inverting the conventional temporal-token representation and treating variates as tokens. This design enhances channel-wise correlation learning and forecasting accuracy while reducing some redundancy in temporal attention computation. However, iTransformer still relies on Transformer-based attention operations whose computational and memory costs increase significantly with longer input sequences, limiting scalability for low-latency deployment scenarios. There is a need for archi-

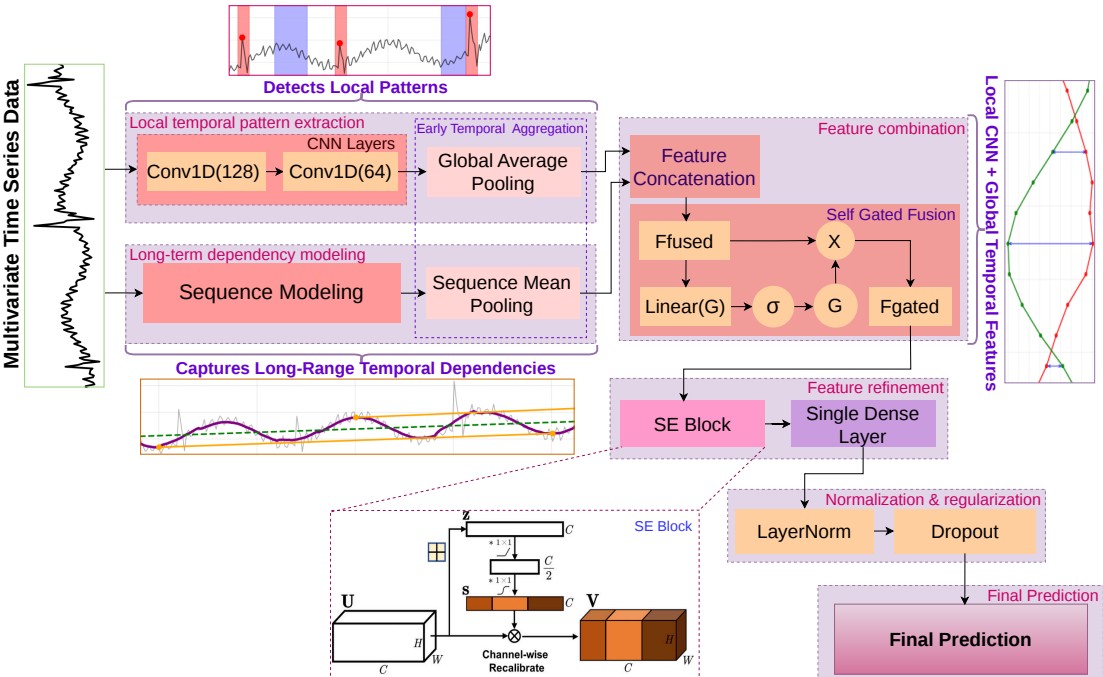

Figure 1: Architecture of the proposed Multi-scale Temporal Network (MSTN).

tectures that are not only general-purpose but also lightweight and efficient. Our work attempts to address this gap.

## 3 Methodology

This section details the MSTN architecture and its methodological innovations, followed by the baseline models, and training configurations.

### 3.1 Proposed Architecture: MSTN

#### 3.1.1 Architecture Overview

The proposed *Multi-scale Temporal Network* (MSTN; Fig. 1) is a hybrid deep-learning architecture designed around a parallel multi-scale modeling paradigm. MSTN processes temporal signals through two coordinated encoding branches, each specialized for complementary aspects of the temporal structure. A key design element is the use of an *early temporal aggregation* (ETA) mechanism, applied directly to the output of the dual encoders, which collapses the full input sequence of length $L$ into a single fixed-dimensional representation $(L \rightarrow 1)$. This operation ensures that all subsequent refinement stages operate in constant time with respect to sequence length, thereby enabling efficient processing of long temporal input.

The network comprises three principal modules. (i) A *multi-scale convolutional branch* captures localized temporal patterns using hierarchically stacked convolutions; its output is aggregated via global average pooling, promoting robustness and translation invariance. (ii) A *sequence modeling branch* captures long-range dependencies, instantiated using either a bidirectional LSTM (MSTN-BiLSTM) or a Transformer encoder (MSTN-Transformer). To maintain computational efficiency, this branch is compressed through sequence mean pooling, yielding a global summary vector. (iii) A *multi-scale fusion and refinement module* integrates the pooled representations from the two branches. The features are concatenated and passed through a SGF mechanism, which adaptively modulates the contributions of each branch. The fused representation is subsequently refined using channel-wise recalibration via SE blocks and is followed by a SDL, enabling

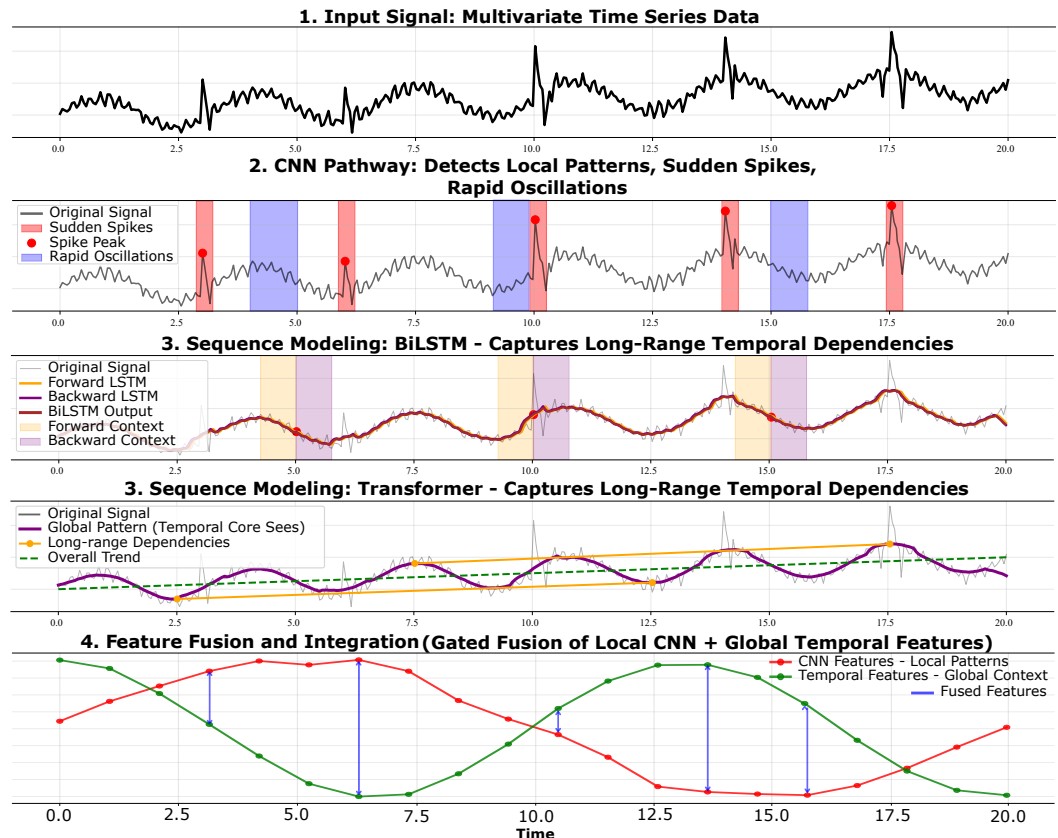

Figure 2: Multi-scale signal processing pipeline of MSTN.

the network to emphasize the most informative feature dimensions. The final representation is normalized, regularized through dropout, and projected to the task-specific output space. For classification tasks, we employ the standard cross-entropy loss defined in Eq. 14. Figure 2 provides an overview of the end-to-end processing pipeline, from parallel feature extraction to multi-scale fusion and refinement.

### 3.1.2 Design Rationale

The MSTN architecture is motivated by specific limitations in existing time series models. Transformer-based approaches excel at global dependency modeling but tend to oversmooth local fluctuations (Nguyen et al., 2023; Noguchi & Kawahara, 2025), while convolutional networks capture fine-grained patterns but have limited receptive fields. Real-world time series are inherently multi-scale, with micro-fluctuations, sudden change events, and long-term trends coexisting—yet most architectures capture only one scale.

MSTN addresses this through a dual-branch design: a multi-scale CNN branch preserves local patterns that attention mechanisms would smooth, while a Transformer/BiLSTM branch models long-range dependencies beyond CNN reach. These complementary representations are dynamically balanced via a SGF mechanism that adaptively weights each branch based on input characteristics, rather than using fixed weights. SE blocks then recalibrate channel-wise feature importance, followed by a SDL that projects the fused representation into a higher-level feature space, enabling nonlinear combinations of the multi-scale temporal features while maintaining computational efficiency. Crucially, ETA collapses the temporal dimension ($L \rightarrow 1$) immediately after encoding, confining all $\mathcal{O}(L)$ and $\mathcal{O}(L^2)$ operations to the initial encoders. Importantly, only the subsequent refinement and prediction stages after ETA operate in $\mathcal{O}(1)$ time; the front-end encoder retains its original complexity ($\mathcal{O}(L^2)$ for Transformer, $\mathcal{O}(L)$ for BiLSTM). This combination yields an architecture that is simultaneously expressive (multi-scale), efficient (constant-time inference for downstream modules),

and adaptable (dynamic fusion of complementary features), directly addressing the core limitations identified in existing approaches.

### 3.1.3 Mathematical Modeling

We consider the following problem: Given multivariate time series samples $X_{1:L} \in \mathbb{R}^{L \times C}$ with a lookback window sequence length $L$ and $C$ variate features, MSTN learns temporal representations through a novel parallel architecture. We detail the complete process below.

Forward Process: The input $X \in \mathbb{R}^{B \times L \times C}$ is processed through dual complementary branches that operate simultaneously. The CNN captures local temporal patterns, while the sequence modeling (implemented as either a Transformer or a BiLSTM) captures long-range temporal dependencies. Their outputs are fused through a sophisticated gating and attention mechanism for enhanced representation learning. The input undergoes successive Conv1D operations to extract multi-scale temporal features, progressively adjusting the feature dimensionality while preserving temporal resolution. The $\text{Conv1D}_k$ convolutional layers employ a hierarchical design in which the first layer with kernel size $k = 7$ extracts basic temporal patterns, and the second layer with kernel size $k = 5$ learns combinations of these patterns into more complex features, with padding to maintain temporal dimensions, and batch normalization stabilizes training. To aggregate these temporal features into a compact representation, we apply global average pooling across the entire sequence, where $X^\top \in \mathbb{R}^{B \times C \times L}$ denotes the channel-first representation obtained by transposing the input along the temporal and variable dimensions.

$$H_{\text{conv}}^{(1)} = \text{ReLU}(\text{BN}(\text{Conv1D}_7(X^T))) \in \mathbb{R}^{B \times 128 \times L} \tag{1}$$

$$H_{\text{conv}}^{(2)} = \text{ReLU}(\text{BN}(\text{Conv1D}_5(H_{\text{conv}}^{(1)}))) \in \mathbb{R}^{B \times 64 \times L} \tag{2}$$

$$\mathbf{z}_{\text{cnn}} = \text{GlobalAvgPool1D}(H_{\text{conv}}^{(2)}) \in \mathbb{R}^{B \times 64} \tag{3}$$

where each of the 64 feature channels is summarized by its average activation over time through temporal pooling. We employ two alternative sequence modeling approaches: MSTN-Transformer and MSTN-BiLSTM. First, the input sequence is projected into a 128-dimensional latent feature space using a linear embedding layer. Specifically, the projected representation is obtained as $X_p = W_p X + b_p$, where $X_p \in \mathbb{R}^{B \times L \times 128}$, and $W_p \in \mathbb{R}^{C \times 128}$ and $b_p \in \mathbb{R}^{128}$ denote the learnable weight matrix and bias of the projection layer that embed the original $C$-dimensional input features into a 128-dimensional latent space. In MSTN-Transformer, the architecture captures long-range dependencies through a *single* Transformer encoder composed of 4 layers with 8 attention heads. The encoder produces contextualized representations for each of the $L$ time steps. To consolidate these temporal representations into a fixed-length global descriptor, we apply sequence mean pooling over the temporal dimension:

$$H_{\text{trans}} = \text{TransformerEnc}(X_p) \in \mathbb{R}^{B \times L \times 128}, \quad \mathbf{z}_{\text{trans}} = \frac{1}{L} \sum_{t=1}^{L} H_{\text{trans},t} \in \mathbb{R}^{B \times 128} \tag{4}$$

In MSTN-BiLSTM, bidirectional sequential dependencies are modeled by processing the projected sequence in both forward and backward directions, enabling the network to learn complex temporal dynamics and long-range contextual patterns:

$$H_{\text{bilstm}} = \text{BiLSTM}(X_p) \in \mathbb{R}^{B \times L \times 128}, \quad \mathbf{z}_{\text{bilstm}} = \frac{1}{L} \sum_{t=1}^{L} H_{\text{bilstm},t} \in \mathbb{R}^{B \times 128} \tag{5}$$

The BiLSTM consists of 2 layers with 64 hidden units per direction, capturing both causal and anti-causal temporal relationships.

The CNN branch utilizes global average pooling, and the sequence modeling branches employ sequence mean pooling. These pooling operations, along with the subsequent reshaping, collectively form the ETA module,

which collapses the temporal dimension $L$ to 1. This critical step ensures that only the subsequent fusion, refinement, and prediction operations after ETA execute at $\mathcal{O}(1)$ inference cost; the encoder complexity remains unchanged ($\mathcal{O}(L^2)$ for Transformer, $\mathcal{O}(L)$ for BiLSTM). This design enables MSTN to achieve constant-time inference after ETA for the refinement and prediction stages only, while preserving both local and global temporal semantics. The parallel features undergo sophisticated fusion through a multi-stage enhancement process that combines the strengths of both branches. For each MSTN variant (MSTN-Transformer or MSTN-BiLSTM):

$$\mathbf{z}_{\text{concat}} = [\mathbf{z}_{\text{cnn}}; \mathbf{z}_{\text{trans/bilstm}}] \in \mathbb{R}^{B \times 192} \tag{6}$$

where $\mathbf{z}_{\text{cnn}} \in \mathbb{R}^{B \times 64}$ and $\mathbf{z}_{\text{trans/bilstm}} \in \mathbb{R}^{B \times 128}$. The architecture then performs adaptive weighting through an SGF mechanism, illustrated in Figure 1, that learns to dynamically balance feature contributions:

$$\mathbf{z}_{\text{fused}} = \mathbf{z}_{\text{concat}} \odot \sigma(W_g \mathbf{z}_{\text{concat}} + b_g) \tag{7}$$

where for both variants: $W_g \in \mathbb{R}^{192 \times 192}$, $b_g \in \mathbb{R}^{192}$. This mechanism learns to emphasize the most relevant features from each branch, creating a harmonious blend of multi-scale temporal features. The fused features are reshaped for efficient single-token processing: $\mathbf{z}_{\text{seq}} = \mathbf{z}_{\text{fused}} \otimes \mathbf{1} \in \mathbb{R}^{B \times 1 \times 192}$. Channel-wise attention then enhances features through an SE block. Given that the temporal dimension is already $L = 1$, the global pooling step within SE is trivially executed: $\mathbf{z}_{\text{se}} = \mathbf{z}_{\text{seq}} \odot \sigma(W_2 \text{ReLU}(W_1 \mathbf{z}_{\text{seq}}))$ where $W_1 \in \mathbb{R}^{24 \times 192}$, $W_2 \in \mathbb{R}^{192 \times 24}$ with a reduction ratio 8. The SE mechanism amplifies informative channels while suppressing less useful ones. Feature dependencies are refined through a SDL. Since the temporal dimension is condensed ($L = 1$), the SDL functions as a sophisticated global feature recalibration layer rather than a temporal mixer, confirming $\mathcal{O}(1)$ complexity:

$$\mathbf{z}_{\text{final}} = \text{Dropout}\left(\text{LayerNorm}\left(W_d \cdot \mathbf{z}_{\text{se}} + b_d\right)\right) \tag{8}$$

where $W_d \in \mathbb{R}^{192 \times 192}$ and $b_d \in \mathbb{R}^{192}$ are the learnable weight matrix and bias of the SDL respectively. Dropout with rate $p = 0.1$ provides regularization, and layer normalization ensures stable training.

The refined features $\mathbf{z}_{\text{final}}$ are then passed to the final task-specific head (imputation, forecasting, or classification). The prediction head consists of a single linear layer that maps the 192-dimensional aggregated feature vector to the task-specific output shape.

Task-Specific Prediction and Loss Functions: We detail the prediction head ($\hat{Y}$ or $\hat{X}$) and the corresponding loss function $\mathcal{L}$ for each of the three tasks. The final representation $\mathbf{z}_{\text{final}}$ is passed through specialized linear heads to map the latent features to the required output space.

Time Series Imputation: The head reconstructs the input sequence $\hat{X}_{1:L} \in \mathbb{R}^{B \times L \times C}$. The objective is to minimize the masked Mean Squared Error (MSE) computed only over masked positions, where $\Omega$ is the binary mask matrix ($\Omega_{i,t,c} = 1$ if the value is masked during training/evaluation):

$$\hat{X}_{1:L} = W_i \mathbf{z}_{\text{final}} + b_i \in \mathbb{R}^{B \times L \times C} \tag{9}$$

$$\mathcal{L}_{\text{imputation}} = \frac{1}{\|\Omega\|_1} \sum_{i=1}^{B} \sum_{t=1}^{L} \sum_{c=1}^{C} \Omega_{i,t,c} \cdot \left(\hat{X}_{i,t,c} - X_{i,t,c}\right)^2 \tag{10}$$

Long-Term Forecasting: The head produces a forecast $\hat{Y}_{1:H} \in \mathbb{R}^{B \times H \times C}$ for a prediction horizon $H$. The objective is to minimize the MSE across the batch, horizon, and variable dimensions:

$$\hat{Y}_{1:H} = W_f \mathbf{z}_{\text{final}} + b_f \in \mathbb{R}^{B \times H \times C} \tag{11}$$

$$\mathcal{L}_{\text{forecast}} = \frac{1}{BHC} \sum_{i=1}^{B} \sum_{t=1}^{H} \sum_{c=1}^{C} \left(\hat{Y}_{i,t,c} - Y_{i,t,c}\right)^2 \tag{12}$$

Table 1: Taxonomy of model architectures by Channel-Interaction strategy. Different state-of-the-art models align with the established Channel Interaction (CI, CD, CP) strategies. MSTN is positioned as a hybrid design (CI+CD) via early temporal aggregation to capture multi-scale dynamics. (Asym.: Asymmetry, Lag.: Lagginess, Pol.: Polarity, Gw.: group-wise, Dyn.: Dynamism, Ms.: Multi-scale).

| Strategy | Mechanism | Characteristic | | | | | | Method |
|---|---|---|---|---|---|---|---|---|
| | | Asym. | Lag. | Pol. | Gw. | Dyn. | Ms. | |
| **CI** | - | - | - | - | - | - | - | PatchTST |
| CI | - | - | - | - | - | - | - | CycleNet |
| CI | - | - | - | - | - | - | - | DLinear |
| CI | - | - | - | - | - | - | - | Timer |
| CI | - | - | - | - | - | - | - | Chronos |
| CI | - | - | - | - | - | - | - | LLM4TS |
| CI | - | - | - | - | - | - | - | Time-LLM |
| CI | - | - | - | - | - | - | - | RevIN |
| **CD** | CNN-based | ✓ | - | - | - | - | - | Informer |
| CD | CNN-based | ✓ | - | - | - | - | - | Autoformer |
| CD | CNN-based | ✓ | - | - | - | - | - | FEDformer |
| CD | CNN-based | ✓ | - | - | - | - | - | TimesNet |
| CD | MLP-based | ✓ | - | - | - | - | - | TSMixer |
| CD | MLP-based | ✓ | - | - | - | - | - | TTM |
| CD | Transformer-based | ✓ | - | - | - | ✓ | - | iTransformer |
| CD | Transformer-based | ✓ | - | - | - | ✓ | - | Crossformer |
| CD | Transformer-based | ✓ | ✓ | - | - | - | - | VCformer |
| CD | Transformer-based | ✓ | ✓ | - | - | ✓ | - | MOIRAI |
| CD | Transformer-based | ✓ | - | - | - | ✓ | - | UniTS |
| CD | GNN-based | - | - | - | ✓ | - | - | GTS |
| CD | GNN-based | ✓ | - | - | - | - | ✓ | MSGNet |
| CD | GNN-based | - | ✓ | - | - | - | - | FourierGNN |
| CD | GNN-based | - | ✓ | - | - | - | - | FC-STGNN |
| CD | GNN-based | ✓ | - | - | - | ✓ | - | TPGNN |
| CD | Linear Channel Mixing (Factorization) | ✓ | - | - | - | - | - | SOFTS |
| CD | Others | ✓ | - | - | - | ✓ | - | C-LoRA |
| **CP** | CNN-based | ✓ | - | - | - | - | - | ModernTCN |
| CP | Transformer-based | ✓ | - | - | ✓ | - | - | DUET |
| CP | Transformer-based | ✓ | - | - | - | ✓ | - | MCformer |
| CP | Transformer-based | ✓ | - | - | ✓ | ✓ | - | DGCformer |
| CP | Transformer-based | - | - | - | - | ✓ | - | CM |
| CP | GNN-based | ✓ | - | - | - | - | - | MTGNN |
| CP | GNN-based | ✓ | - | ✓ | - | - | - | CrossGNN |
| CP | GNN-based | ✓ | - | - | ✓ | - | - | WaveForM |
| CP | GNN-based | - | - | - | - | ✓ | - | MTSF-DG |
| CP | GNN-based | ✓ | - | - | ✓ | - | - | ReMo |
| CP | GNN-based | ✓ | - | - | ✓ | ✓ | ✓ | Ada-MSHyper |
| CP | Others | ✓ | ✓ | - | - | - | - | LIFT |
| CP | Others | ✓ | - | - | ✓ | - | - | CCM |
| **CI + CD** | Parallel CNN (CI) and BiLSTM/Transformer(CD) based Dual Encoder + ETA | ✓ | - | - | - | ✓ | ✓ | **MSTN** |

Time Series Classification: The head computes class probabilities $P \in \mathbb{R}^{B \times K}$ over $K$ target classes. The model is optimized using the canonical Cross-Entropy (CE) Loss:

$$P = \text{Softmax}(W_c \mathbf{z}_{\text{final}} + b_c) \in \mathbb{R}^{B \times K} \tag{13}$$

$$\mathcal{L}_{\text{classify}} = -\frac{1}{B} \sum_{i=1}^{B} \sum_{k=1}^{K} Y_{i,k} \log(P_{i,k}) \tag{14}$$

where $Y$ is the one-hot encoded true label. This architectural modularity allows the MSTN framework to be universally applied to diverse multivariate time series tasks without structural modification.

## 3.2 How is MSTN different from prior works?

The established field of multivariate time–series forecasting is conventionally organized around three channel interaction strategies—CI, CD, and CP approaches (Qiu et al., 2025)—as summarized in Table 1. These strategies specify how a model encodes and exploits inter-variable dependencies across the $C$ channels of a multivariate signal.

Table 2: Comparison of architectural characteristics. Evaluation of the MSTN framework against the CI, CD, and CP Channel Interaction strategies. MSTN achieves high capacity while neutralizing the $\mathcal{O}(L^2)$ computational bottleneck.

| Dimension | CI | CD | CP | (CI+CD) MSTN | Rationale |
|---|---|---|---|---|---|
| Efficiency | High | Low | Moderate | High | ETA enables $\mathcal{O}(1)$ refinement after initial encoding. |
| Robustness | High | Low | Moderate | High | Dual-encoder with SGF provides complementary feature stability. |
| Generalizability | Low | Moderate | High | High | Achieved via a single, aggregated feature vector ($\mathbf{z}_{\text{final}}$) that supports multi-task learning (Forecasting, Imputation, Classification). |
| Capacity | Low | High | Moderate | High | Transformer/BiLSTM path captures global dependencies. |
| Ease of Implementation | High | Moderate | Low | Moderate | Relies on standard components (CNN, Transformer/BiLSTM, SGF, SE, SDL) rather than complex, specialized graph or recursive structures. |

**CI.** CI methods (e.g., DLinear, PatchTST) process each variable independently using lightweight temporal operators such as linear layers or multilayer perceptrons. Their advantage lies in linear channel cost $\mathcal{O}(C)$ and efficient temporal complexity $\mathcal{O}(L)$, though this comes at the expense of discarding cross-channel structure.

**CD.** CD approaches (e.g., iTransformer) explicitly model inter-variable correlation through dense attention or graph-based mechanisms. While these models offer comprehensive feature coupling, they incur high computational cost—typically $\mathcal{O}(C^2)$ or $\mathcal{O}(L^2)$—which limits their scalability and latency characteristics.

**CP.** CP models (e.g., MCformer, MTGNN) map the channel dimension into a low-rank latent space before applying interaction, thereby avoiding the full $\mathcal{O}(C^2)$ cost of CD models. Although channel-efficient, these models often impose simplified temporal assumptions, reducing their ability to capture multi-scale dynamics.

By combining the complementary strengths of CI and CD models with a novel temporal aggregation mechanism, MSTN delivers strong performance across forecasting, imputation, and classification tasks while maintaining the low-latency characteristics.

### 3.2.1 Core Innovation: Dual-Path Design and ETA

The MSTN framework integrates a dual-path temporal encoding architecture with the ETA mechanism, resulting in a principled hybrid approach that reconciles the core trade-offs among CI-, CD-, and CP-based modeling strategies as shown in Table 2. Its design is centered on two architectural innovations that jointly enhance representational fidelity and computational efficiency:

1. **Dual-branch temporal encoding.** In contrast to recent CP-oriented architectures such as MCformer (Han et al., 2024), which priorities efficiency at the expense of temporal expressiveness, MSTN explicitly preserves multi-scale temporal structure. It deploys a parallel encoding strategy in which a global sequence-modeling branch (Transformer or BiLSTM; CD strategy) captures long-range temporal dependencies, while a lightweight convolutional branch (CI strategy; $\mathcal{O}(L)$) extracts fine-grained local temporal patterns. This duality directly addresses the multi-scale nature of real-world temporal signals, offering coverage beyond what is typically afforded by efficient CP-based methods.

2. **Efficiency through ETA: collapsing $\mathcal{O}(L^2)$ to $\mathcal{O}(1)$.** MSTN applies the ETA mechanism immediately after the high-capacity encoders, collapsing the temporal dimension ($L \to 1$) via learned sequence aggregation. By performing the computationally intensive operations (e.g., $\mathcal{O}(L^2)$ Transformer self-attention) *before* this aggregation step, the subsequent refinement layers—including feature fusion, SE recalibration, SDL, and prediction modules—operate with a fixed $\mathcal{O}(1)$ cost with respect to $L$. This explicitly reshapes the model's complexity profile, enabling MSTN to maintain the representational power of CD architectures while achieving substantially improved inference efficiency. Unlike conventional temporal pooling, hierarchical downsampling, or token-merging strategies, ETA is not introduced as a representational shortcut, but as a principled reordering of computation. Prior approaches typically aggregate temporal information either progressively across

layers or at the final prediction stage, implicitly coupling representational capacity with inference-time complexity. In contrast, ETA explicitly allows high-capacity temporal modeling to operate on the full input sequence, while enforcing an early collapse of the temporal dimension before downstream fusion and refinement.

# 4 Experiments

This section presents a comprehensive evaluation of the proposed MSTN model. We demonstrate its performance on three time-series tasks: (1) imputation, (2) long-term forecasting, (3) classification. Additionally, we also present the generalizability study on standard time-series benchmark datasets.

## 4.1 Baselines

This section presents the baselines used to assess MSTN on four time series tasks: (1) imputation, (2) long-term forecasting, (3) classification, and (4) generalizability study.

### 4.1.1 Imputation

Following the TimesNet (Wu et al., 2023) protocol, we evaluate performance under the Missing Completely at Random (MCAR) scenario with random masking ratios of $\{12.5\%, 25\%, 37.5\%, 50\%\}$. Performance is quantified using Mean Squared Error (MSE) and Mean Absolute Error (MAE). The evaluation includes recent SOTA baselines such as GPT2(3) (Zhou et al., 2023), TimesNet (Wu et al., 2023), PatchTST (Nie et al., 2023), LightTS (Zhang et al., 2022), and DLinear (Zeng et al., 2022).

### 4.1.2 Long-Term Forecasting

The proposed MSTN model is evaluated for the long-term time series forecasting task. We evaluated the proposed MSTN model for forecasting on four public subsets of California traffic network data from the Performance Measurement System (PEMS), namely PEMS03, PEMS04, PEMS07, and PEMS08, with prediction horizons $H \in \{12, 24, 48, 96\}$, following the protocol of iTransformer (Liu et al., 2024). We compare our model against state-of-the-art baselines including iTransformer (Liu et al., 2024), PatchTST (Nie et al., 2023), and TSMixer (Chen et al., 2023).

### 4.1.3 Classification

The performance of classification tasks is evaluated using the standard metric of classification accuracy. The evaluation compares with recent SOTA baselines such as GPT2(6) (Zhou et al., 2023), TimesNet (Wu et al., 2023), DLinear (Zeng et al., 2022), ETSFormer (Woo et al., 2022), FEDformer (Zhou et al., 2022), and Informer (Zhou et al., 2021).

### 4.1.4 Generalizability Study

To evaluate the generalizability of the proposed MSTN framework across domains, we benchmarked it against TimesNet (Wu et al., 2023), PatchTST (Nie et al., 2023), and prior work models from the respective literature (e.g., Random Forest (RF), Gradient Boosting (GB), Decision Tree (DT), k-means, Naive Bayes (NB), k-Nearest Neighbors (k-NN), and Stacked Autoencoder (SAE)). All evaluations utilized a consistent five-fold cross-validation framework on seven publicly available international datasets spanning healthcare, activity recognition, agricultural technology, and industrial monitoring.

## 4.2 Experimental Setup

### 4.2.1 Data Preprocessing and Evaluation Protocol

Our preprocessing and splits follow established practices from prior work (Zhou et al., 2023; Liu et al., 2024; Wu et al., 2023; Nie et al., 2023). All datasets are split chronologically to preserve temporal causality. For ETT datasets (ETTh1, ETTh2, ETTm1, ETTm2), we use a 60%/20%/20% train/validation/test

split and for other datasets such as ECL, weather, we adopt a 70%/10%/20% split for imputation tasks. The raw time series is partitioned before constructing sliding windows to avoid temporal overlap across splits. All features are normalized using train-only statistics, where the mean and standard deviation are computed from the training set and applied to the validation and test sets. For imputation tasks, evaluation metrics (MSE and MAE) are computed on normalized predictions under random masking ratios of $\{12.5\%, 25\%, 37.5\%, 50\%\}$, consistent with established benchmarks (Nie et al., 2023; Wu et al., 2023; Liu et al., 2024). Furthermore, we evaluated the proposed model on four PEMS datasets (PEMS03, PEMS04, PEMS07, PEMS08) for long-term forecasting, following the protocol of iTransformer (Liu et al., 2024). We use a lookback window of $L = 96$ and prediction horizons $H \in \{12, 24, 48, 96\}$. The data is split chronologically into 60% training, 20% validation, and 20% testing. Metrics (MSE and MAE) are reported on the normalized scale, consistent with the prior benchmarks. We evaluate MSTN on ten multivariate time series classification datasets from the University of East Anglia (UEA) archive (Bagnall et al., 2018), following the standard protocol of GPT2(6) (Zhou et al., 2023) and TimesNet (Wu et al., 2023) using official train/test splits. Performance is evaluated using classification accuracy (%). Metrics are reported on the original scale, consistent with prior benchmarks. The generalizability study used seven international cross-domain datasets: Rodegast, Boubezoul, UCI-HAR, PAMAP2, ActBeCalf, MetroPT3, and NASA. A chronological 80/10/10 train/validation/test split was adopted for all models, including MSTN, TimesNet, and PatchTST, to prevent temporal leakage and ensure consistent evaluation. Accuracy is reported for classification tasks and the root mean square error (RMSE) is reported for the NASA dataset.

### 4.2.2 Training and Model Configuration

The proposed MSTN model is implemented in Python 3.13.1 using PyTorch 2.7.1 and trained on an NVIDIA DGX A100 server equipped with $8 \times$ NVIDIA A100-SXM4 GPUs (40 GB HBM2e VRAM each), using CUDA 12.3 and driver version 535.54.03. The architecture processes variable-length sequences through either a sequence modeling core (BiLSTM hidden units: 64 per direction (128 total)) or a Transformer (4 layers, 8 attention heads), combined with a CNN branch ($128 \rightarrow 64$ filters). Training is performed using the AdamW (Kingma & Ba, 2017) optimizer with a learning rate of $1 \times 10^{-4}$, batch size of 64, and task-specific loss functions. For forecasting and imputation tasks, Mean Squared Error (MSE) loss is used, while for classification tasks, Cross-Entropy Loss is employed. For imputation, a masked MSE loss is applied, where the loss is computed only over randomly masked time points. All experiments follow the preprocessing protocol described in Section 4.2.1, including train-only standardization (Z-score normalization). For forecasting and imputation tasks, a fixed input lookback window of $L = 96$ is used across all prediction horizons to ensure consistency with prior benchmarks. All models are trained for up to 50 epochs with early stopping based on validation performance. Hyperparameters are tuned following standard practice in time-series forecasting literature (Madhusudhanan et al., 2024; Fristiana et al., 2024). To ensure robustness and reproducibility, all experiments are repeated with 5 different random seeds and results are reported as mean $\pm$ standard deviation.

### 4.2.3 Evaluation Metrics

The performance of imputation and long-term forecasting tasks is evaluated using Mean Squared Error (MSE) and Mean Absolute Error (MAE), computed over all prediction horizons and averaged across all variables. For classification tasks, we report Accuracy as the primary metric, following standard practice in the literature (e.g., GPT-2 (Zhou et al., 2023), iTransformer (Liu et al., 2024), TimesNet (Wu et al., 2023)).

### 4.3 Benchmark Datasets

We evaluate the proposed method across three fundamental time series tasks: imputation, long-term forecasting, and classification using established public benchmark datasets. Table 3 provides a detailed overview of datasets. Missing value imputation is performed under a MCAR setting, with masking ratios ranging from 12.5%, 25%, 37.5% and 50%. Benchmarks include ETTh1, ETTh2, ETTm1, ETTm2, Electricity, and Weather datasets (Zhou et al., 2021; Trindade, 2015; Köllé, 2025). Imputation results are quantified using MSE and MAE over sequence lengths of 96 time steps.

Table 3: Benchmark Datasets for Imputation, Forecasting and Classification (10 UEA).

| Task | Datasets | Features | Series Length | Dataset Size | Information |
|---|---|---|---|---|---|
| **Imputation** | ETTh1 | 7 | 96 | (8,545, 2,881, 2,881) | Electricity |
| | ETTh2 | 7 | 96 | (8,545, 2,881, 2,881) | Electricity |
| | ETTm1 | 7 | 96 | (34,465, 11,521, 11,521) | Electricity |
| | ETTm2 | 7 | 96 | (34,465, 11,521, 11,521) | Electricity |
| | Electricity | 321 | 96 | (18,317, 2,633, 5,261) | Electricity |
| | Weather | 21 | 96 | (36,792, 5,271, 10,540) | Weather |
| **Forecasting** | PEMS03 | 358 | {12,24,48,96} | (15,724, 5,241, 5,243) | Transportation |
| | PEMS04 | 307 | {12,24,48,96} | (10,195, 3,398, 3,399) | Transportation |
| | PEMS07 | 883 | {12,24,48,96} | (16,934, 5,644, 5,646) | Transportation |
| | PEMS08 | 170 | {12,24,48,96} | (10,713, 3,571, 3,572) | Transportation |
| **Classification** | EthanolConcentration | 3 | 1,751 | (261, 0, 263) | Alcohol Industry |
| | FaceDetection | 144 | 62 | (5,890, 0, 3,524) | Face |
| | Handwriting | 3 | 152 | (150, 0, 850) | Handwriting |
| | Heartbeat | 61 | 405 | (204, 0, 205) | Heart Beat |
| | JapaneseVowels | 12 | 29 | (270, 0, 370) | Voice |
| | PEMS-SF | 963 | 144 | (267, 0, 173) | Transportation |
| | SelfRegulationSCP1 | 6 | 896 | (268, 0, 293) | Health |
| | SelfRegulationSCP2 | 7 | 1,152 | (200, 0, 180) | Health |
| | SpokenArabicDigits | 13 | 93 | (6,599, 0, 2,199) | Voice |
| | UWaveGestureLibrary | 3 | 315 | (120, 0, 320) | Gesture |

Long-term forecasting performance is assessed on four PEMS traffic datasets (PEMS03, PEMS04, PEMS07, PEMS08), following the protocols of iTransformer (Liu et al., 2024), PatchTST (Nie et al., 2023), and TSMixer (Chen et al., 2023). These datasets are extensively used for time-series benchmarking and are publicly available (Lin, 2025; Liu et al., 2022a; Chen & data, 2025; Wu et al., 2022). Long-term forecasting performance is evaluated using MSE and MAE.

The multivariate time series classification is evaluated on ten diverse datasets from the University of East Anglia (UEA) archive (Bagnall et al., 2018), which includes applications from healthcare to activity recognition. To rigorously assess cross-domain transferability, we utilize seven publicly available international datasets spanning diverse applications: Human Safety/Healthcare (Fall Event (Boubezoul et al., 2020), Risky Driving (Rodegast et al., 2024a)); Human Activity Recognition (HAR) (UCI-HAR (Reyes-Ortiz et al., 2013), PAMAP2 (Reiss, 2012)); Agricultural Technology/Welfare (ActBe-Calf (Dissanayake et al., 2025)); and Industrial Monitoring/Predictive Maintenance (MetroPT-3 (Davari et al., 2021a), NASA Turbofan Engine (Saxena & Goebel, 2008)).

## 4.4 Imputation Results

Imputation is the task of filling in missing values in a dataset. In real-world time series data, sensors can fail or data can be corrupted, which may create gaps in the data. This task tests the ability of the model to intelligently reconstruct missing data based on the surrounding context, which is vital for maintaining data quality.

Following the TimesNet (Wu et al., 2023) protocol, we evaluate imputation performance on electricity and weather domain datasets, including ETT (Zhou et al., 2021), ECL (Trindade, 2015), and Weather (Köllé, 2025). We evaluate performance under the MCAR scenario with random masking ratios of $\{12.5\%, 25\%, 37.5\%, 50\%\}$. During training, we randomly mask the specified percentage of time points. The model reconstructs the complete sequence, but the loss is computed only on masked positions to ensure the model learns to impute missing values rather than memorize observed ones. At test time, we apply the same masking protocol and evaluate imputation performance by computing MSE and MAE exclusively on the masked positions. This ensures that the model is assessed solely on its imputation capability. Performance is quantified using MSE and MAE. The evaluation includes recent SOTA baselines such as GPT2(3) (Zhou et al., 2023), TimesNet (Wu et al., 2023), PatchTST (Nie et al., 2023), LightTS (Zhang et al., 2022), and DLinear (Zeng et al., 2022).

Table 4: Imputation Task Results Comparison. We randomly mask 12.5%, 25%, 37.5%, and 50% time points to compare model performance under different missing degrees. Red/Blue: First/Second ranks. We evaluate each model five times and report the mean (standard deviation); results marked with underline are not significantly worse than the first rank (Wilcoxon signed-rank test with Holm–Bonferroni correction, $\alpha = 0.05$). Example: 0.010(0.001) means $0.010 \pm 0.001$. Tra.: Transformer, BiL.: BiLSTM, FED.: FEDformer.

| Models | MSTN BiL. (Ours) | | MSTN-Tra. (Ours) | | GPT2(3) | | TimesNet | | PatchTST | | LightTS | | DLinear | | FED. | | Stationary | |
|---|---|---|---|---|---|---|---|---|---|---|---|---|---|---|---|---|---|---|
| Mask R. | MSE | MAE | MSE | MAE | MSE | MAE | MSE | MAE | MSE | MAE | MSE | MAE | MSE | MAE | MSE | MAE | MSE | MAE |
| **ETTh1** | | | | | | | | | | | | | | | | | | |
| 12.5% | 0.052(0.00) | 0.059(0.00) | 0.039 | 0.053 | 0.043 | 0.140 | 0.057 | 0.159 | 0.093 | 0.201 | 0.240 | 0.345 | 0.151 | 0.267 | 0.070 | 0.190 | 0.060 | 0.165 |
| 25% | 0.099(0.00) | 0.116(0.00) | 0.079 | 0.107 | 0.054 | 0.156 | 0.069 | 0.178 | 0.107 | 0.217 | 0.265 | 0.364 | 0.180 | 0.292 | 0.106 | 0.236 | 0.080 | 0.189 |
| 37.5% | 0.143(0.00) | 0.169(0.00) | 0.122 | 0.162 | 0.072 | 0.180 | 0.084 | 0.196 | 0.120 | 0.230 | 0.296 | 0.382 | 0.215 | 0.318 | 0.124 | 0.258 | 0.102 | 0.212 |
| 50% | 0.187(0.01) | 0.224(0.01) | 0.168 | 0.219 | 0.107 | 0.216 | 0.102 | 0.215 | 0.141 | 0.248 | 0.334 | 0.404 | 0.257 | 0.347 | 0.165 | 0.299 | 0.133 | 0.240 |
| **Avg** | 0.120 | 0.142 | 0.102 | 0.135 | 0.069 | 0.173 | 0.078 | 0.187 | 0.115 | 0.224 | 0.284 | 0.373 | 0.201 | 0.306 | 0.117 | 0.246 | 0.094 | 0.201 |
| **ETTh2** | | | | | | | | | | | | | | | | | | |
| 12.5% | 0.088(0.00) | 0.083(0.00) | 0.079 | 0.081 | 0.039 | 0.125 | 0.040 | 0.130 | 0.057 | 0.152 | 0.101 | 0.231 | 0.100 | 0.216 | 0.095 | 0.212 | 0.042 | 0.133 |
| 25% | 0.169(0.00) | 0.139(0.00) | 0.154 | 0.157 | 0.044 | 0.135 | 0.046 | 0.141 | 0.061 | 0.158 | 0.115 | 0.246 | 0.127 | 0.247 | 0.137 | 0.258 | 0.049 | 0.147 |
| 37.5% | 0.242(0.01) | 0.229(0.01) | 0.240 | 0.234 | 0.051 | 0.147 | 0.052 | 0.151 | 0.067 | 0.166 | 0.126 | 0.257 | 0.158 | 0.276 | 0.187 | 0.304 | 0.056 | 0.158 |
| 50% | 0.305(0.01) | 0.296(0.01) | 0.364 | 0.328 | 0.059 | 0.158 | 0.060 | 0.162 | 0.073 | 0.174 | 0.136 | 0.268 | 0.183 | 0.299 | 0.232 | 0.341 | 0.065 | 0.170 |
| **Avg** | 0.201 | 0.187 | 0.247 | 0.215 | 0.048 | 0.141 | 0.049 | 0.146 | 0.065 | 0.163 | 0.119 | 0.250 | 0.142 | 0.259 | 0.163 | 0.279 | 0.053 | 0.152 |
| **ETTm1** | | | | | | | | | | | | | | | | | | |
| 12.5% | 0.010(0.00) | 0.028(0.00) | 0.012 | 0.032 | 0.017 | 0.085 | 0.019 | 0.092 | 0.041 | 0.130 | 0.075 | 0.180 | 0.058 | 0.162 | 0.035 | 0.135 | 0.026 | 0.107 |
| 25% | 0.021(0.00) | 0.045(0.00) | 0.022 | 0.055 | 0.022 | 0.096 | 0.023 | 0.101 | 0.044 | 0.135 | 0.093 | 0.206 | 0.080 | 0.193 | 0.052 | 0.166 | 0.032 | 0.119 |
| 37.5% | 0.027(0.00) | 0.076(0.00) | 0.030 | 0.085 | 0.029 | 0.111 | 0.029 | 0.111 | 0.049 | 0.143 | 0.113 | 0.231 | 0.103 | 0.219 | 0.069 | 0.191 | 0.039 | 0.131 |
| 50% | 0.039(0.00) | 0.111(0.00) | 0.044 | 0.115 | 0.040 | 0.128 | 0.036 | 0.124 | 0.055 | 0.151 | 0.134 | 0.255 | 0.132 | 0.248 | 0.089 | 0.218 | 0.047 | 0.145 |
| **Avg** | 0.024 | 0.065 | 0.027 | 0.072 | 0.028 | 0.105 | 0.027 | 0.107 | 0.047 | 0.140 | 0.104 | 0.218 | 0.093 | 0.206 | 0.062 | 0.177 | 0.036 | 0.126 |
| **ETTm2** | | | | | | | | | | | | | | | | | | |
| 12.5% | 0.044(0.00) | 0.054(0.00) | 0.080 | 0.076 | 0.017 | 0.076 | 0.018 | 0.080 | 0.026 | 0.094 | 0.034 | 0.127 | 0.062 | 0.166 | 0.056 | 0.159 | 0.021 | 0.088 |
| 25% | 0.091(0.00) | 0.110(0.00) | 0.141 | 0.141 | 0.020 | 0.080 | 0.020 | 0.085 | 0.028 | 0.099 | 0.042 | 0.143 | 0.085 | 0.196 | 0.080 | 0.195 | 0.024 | 0.096 |
| 37.5% | 0.140(0.00) | 0.166(0.00) | 0.182 | 0.194 | 0.022 | 0.087 | 0.023 | 0.091 | 0.030 | 0.104 | 0.051 | 0.159 | 0.106 | 0.222 | 0.110 | 0.231 | 0.027 | 0.103 |
| 50% | 0.187(0.01) | 0.224(0.01) | 0.201 | 0.234 | 0.025 | 0.095 | 0.026 | 0.098 | 0.034 | 0.110 | 0.059 | 0.174 | 0.131 | 0.247 | 0.156 | 0.276 | 0.030 | 0.108 |
| **Avg** | 0.116 | 0.139 | 0.151 | 0.161 | 0.021 | 0.084 | 0.022 | 0.088 | 0.029 | 0.102 | 0.046 | 0.151 | 0.096 | 0.208 | 0.101 | 0.215 | 0.026 | 0.099 |
| **ECL** | | | | | | | | | | | | | | | | | | |
| 12.5% | 0.030(0.00) | 0.042(0.00) | 0.035 | 0.047 | 0.080 | 0.194 | 0.085 | 0.202 | 0.055 | 0.160 | 0.102 | 0.229 | 0.092 | 0.214 | 0.107 | 0.237 | 0.093 | 0.210 |
| 25% | 0.060(0.00) | 0.084(0.00) | 0.068 | 0.091 | 0.087 | 0.203 | 0.089 | 0.206 | 0.065 | 0.175 | 0.121 | 0.252 | 0.118 | 0.247 | 0.120 | 0.251 | 0.097 | 0.214 |
| 37.5% | 0.083(0.00) | 0.125(0.00) | 0.101 | 0.136 | 0.094 | 0.211 | 0.094 | 0.213 | 0.076 | 0.189 | 0.141 | 0.273 | 0.144 | 0.276 | 0.136 | 0.266 | 0.102 | 0.220 |
| 50% | 0.110(0.00) | 0.165(0.00) | 0.199 | 0.229 | 0.101 | 0.220 | 0.100 | 0.221 | 0.091 | 0.208 | 0.160 | 0.293 | 0.175 | 0.305 | 0.158 | 0.284 | 0.108 | 0.228 |
| **Avg** | 0.071 | 0.104 | 0.101 | 0.126 | 0.090 | 0.207 | 0.092 | 0.210 | 0.072 | 0.183 | 0.131 | 0.262 | 0.132 | 0.260 | 0.130 | 0.259 | 0.100 | 0.218 |
| **Weather** | | | | | | | | | | | | | | | | | | |
| 12.5% | 0.009(0.00) | 0.018(0.00) | 0.007 | 0.014 | 0.026 | 0.049 | 0.025 | 0.045 | 0.029 | 0.049 | 0.047 | 0.101 | 0.039 | 0.084 | 0.041 | 0.107 | 0.027 | 0.051 |
| 25% | 0.018(0.00) | 0.035(0.00) | 0.014 | 0.028 | 0.028 | 0.052 | 0.029 | 0.052 | 0.031 | 0.053 | 0.052 | 0.111 | 0.048 | 0.103 | 0.064 | 0.163 | 0.029 | 0.056 |
| 37.5% | 0.027(0.00) | 0.052(0.00) | 0.021 | 0.042 | 0.033 | 0.060 | 0.031 | 0.057 | 0.035 | 0.058 | 0.058 | 0.121 | 0.057 | 0.117 | 0.107 | 0.229 | 0.033 | 0.062 |
| 50% | 0.036(0.00) | 0.070(0.00) | 0.029 | 0.058 | 0.037 | 0.065 | 0.034 | 0.062 | 0.038 | 0.063 | 0.065 | 0.133 | 0.066 | 0.134 | 0.183 | 0.312 | 0.037 | 0.068 |
| **Avg** | 0.023 | 0.044 | 0.018 | 0.036 | 0.031 | 0.056 | 0.030 | 0.054 | 0.033 | 0.056 | 0.055 | 0.117 | 0.052 | 0.110 | 0.099 | 0.203 | 0.032 | 0.059 |

As presented in Table 4, MSTN-BiLSTM achieves improved average performance on ETTm1 with average MSE of 0.024, outperforming all baselines. On ECL, MSTN-BiLSTM achieves the better average MSE of 0.071, surpassing TimesNet (0.092) and GPT2(3) (0.090). On Weather, MSTN-Transformer achieves the improved overall performance with an average MSE of 0.018, second rank MSTN-BiLSTM (0.023) and outperforming TimesNet (0.030). On ETTh1 with 12.5% masking, MSTN-Transformer achieves MSE of 0.039, outperforming TimesNet (0.057) by 31.6%. On the challenging Weather dataset at 12.5% masking, MSTN-Transformer achieves an MSE of 0.007, surpassing GPT2(3) (0.026) and TimesNet (0.025). These results validate the effectiveness of MSTN's multi-scale design for robust missing value reconstruction under varying missingness levels.

Figure 3 visualizes imputation results on real-world weather data with a 50% mask ratio across all evaluated methods—MSTN-Transformer, GPT2(3), TimesNet, PatchTST, LightTS, DLinear and FEDformer. The results show that MSTN-Transformer closely aligns with the ground truth temporal dynamics. Most baselines follow the general shape of the ground truth, but struggle with finer-grained temporal dynamics. In contrast, MSTN-Transformer consistently tracks both smooth trends and sharp fluctuations, indicating its effectiveness in capturing multi-scale temporal dependencies.

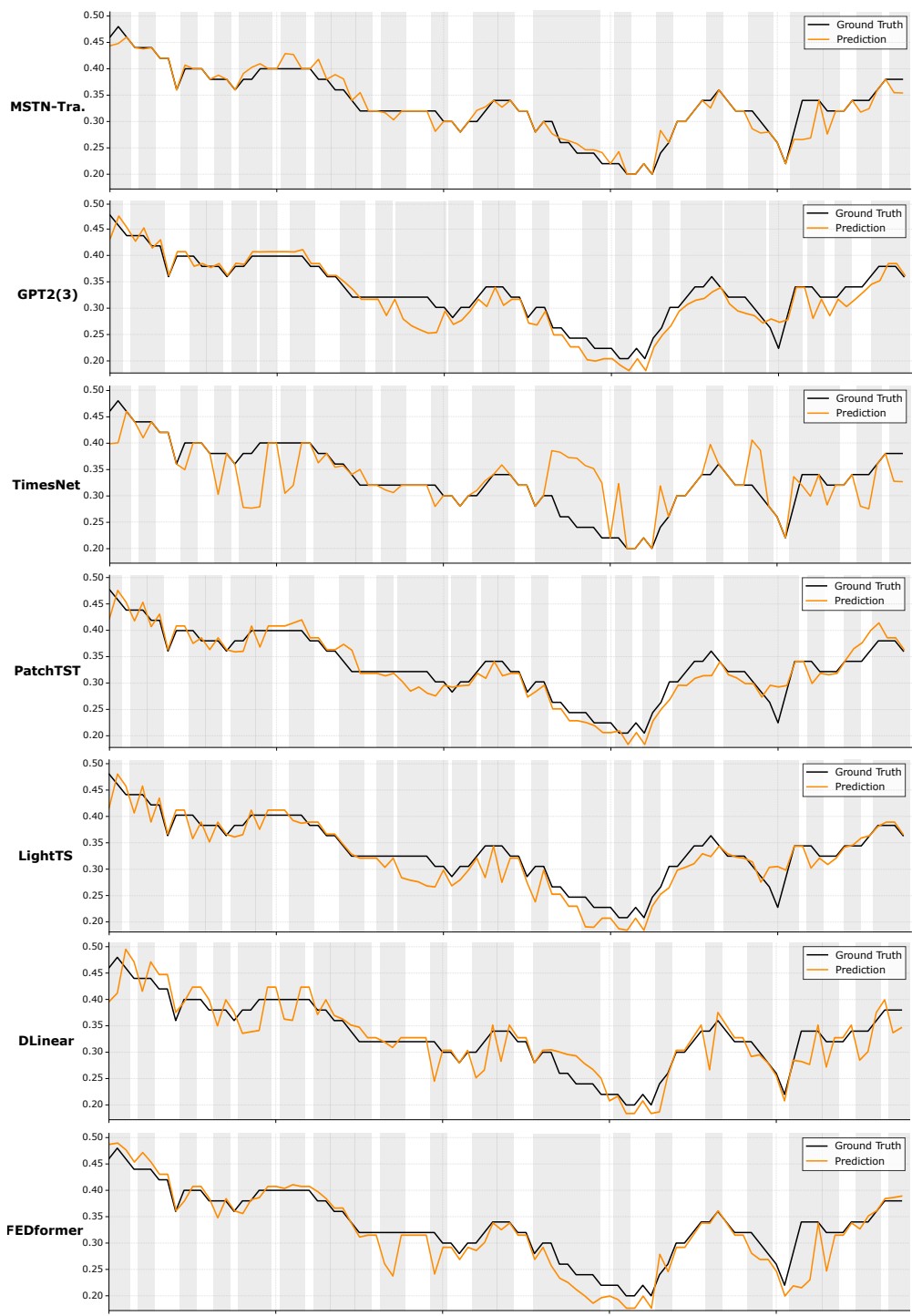

Figure 3: Visualization of weather imputation results under 50% mask ratio. The black lines represent ground truth and the orange lines represent predicted values.

## 4.5 Long-Term Forecasting Results

Long-term forecasting involves predicting future values over extended horizons, ranging from hours to days ahead. Following the protocol of iTransformer (Liu et al., 2024), we evaluate MSTN on four PEMS traffic

Table 5: Long-term forecasting evaluation results on PEMS datasets. Prediction horizons $H \in \{12, 24, 48, 96\}$. Red/Blue: First/Second ranks. We evaluate each model five times and report the mean (standard deviation); results marked with underline are not significantly worse than the first rank (Wilcoxon signed-rank test with Holm–Bonferroni correction, $\alpha = 0.05$).

| Dataset/H | MSTN-Trans. (Ours) | | MSTN-BiL. (Ours) | | iTransformer | | ModernTCN | | PatchTST | | TSMixer | | Autoformer | |
|---|---|---|---|---|---|---|---|---|---|---|---|---|---|---|
| H-Metric | MSE | MAE | MSE | MAE | MSE | MAE | MSE | MAE | MSE | MAE | MSE | MAE | MSE | MAE |
| **PEMS03** | | | | | | | | | | | | | | |
| 12 | 0.131(0.00) | 0.237(0.00) | 0.141 | 0.249 | 0.069 | 0.175 | 0.112 | 0.221 | 0.079 | 0.187 | 0.075 | 0.187 | 0.277 | 0.387 |
| 24 | 0.135(0.00) | 0.241(0.00) | 0.151 | 0.259 | 0.098 | 0.209 | 0.173 | 0.281 | 0.124 | 0.235 | 0.113 | 0.238 | 0.422 | 0.466 |
| 48 | **0.174(0.01)** | **0.268(0.01)** | 0.178 | 0.280 | 0.448 | 0.416 | 0.307 | 0.395 | 0.223 | 0.319 | 0.195 | 0.320 | 0.806 | 0.679 |
| 96 | **0.182(0.01)** | **0.281(0.01)** | 0.197 | 0.299 | 1.215 | 0.831 | 1.041 | 0.779 | 0.368 | 0.425 | 0.266 | 0.380 | 0.710 | 0.634 |
| **Avg** | **0.156** | **0.257** | 0.167 | 0.272 | 0.458 | 0.408 | 0.408 | 0.419 | 0.199 | 0.291 | 0.162 | 0.281 | 0.554 | 0.542 |
| **PEMS04** | | | | | | | | | | | | | | |
| 12 | 0.105(0.00) | 0.210(0.00) | 0.114 | 0.222 | 0.081 | 0.188 | 0.132 | 0.245 | 0.101 | 0.209 | 0.085 | 0.195 | 0.562 | 0.577 |
| 24 | **0.111(0.00)** | **0.217(0.00)** | 0.120 | 0.228 | 0.124 | 0.232 | 0.244 | 0.338 | 0.161 | 0.267 | 0.112 | 0.228 | 0.637 | 0.617 |
| 48 | **0.119(0.00)** | **0.229(0.01)** | 0.127 | 0.239 | 0.135 | 0.248 | 0.452 | 0.482 | 0.294 | 0.369 | 0.159 | 0.278 | 1.002 | 0.775 |
| 96 | **0.126(0.01)** | **0.235(0.01)** | 0.137 | 0.247 | 0.169 | 0.280 | 1.127 | 0.818 | 0.507 | 0.505 | 0.190 | 0.313 | 0.853 | 0.708 |
| **Avg** | **0.115** | **0.226** | 0.125 | 0.240 | 0.127 | 0.237 | 0.488 | 0.471 | 0.266 | 0.338 | 0.136 | 0.254 | 0.764 | 0.669 |
| **PEMS07** | | | | | | | | | | | | | | |
| 12 | 0.131(0.00) | 0.221(0.00) | 0.179 | 0.254 | 0.066 | 0.164 | 0.085 | 0.196 | 0.076 | 0.180 | 0.070 | 0.177 | 0.201 | 0.330 |
| 24 | 0.142(0.01) | 0.232(0.01) | 0.183 | 0.260 | 0.087 | 0.190 | 0.127 | 0.245 | 0.127 | 0.234 | 0.105 | 0.221 | 0.304 | 0.402 |
| 48 | **0.155(0.01)** | **0.241(0.01)** | 0.189 | 0.265 | 0.892 | 0.764 | 0.267 | 0.380 | 0.238 | 0.325 | 0.157 | 0.265 | 0.422 | 0.472 |
| 96 | **0.168(0.02)** | **0.252(0.02)** | 0.196 | 0.273 | 0.972 | 0.789 | 0.736 | 0.673 | 0.394 | 0.432 | 0.268 | 0.342 | 0.519 | 0.546 |
| **Avg** | **0.149** | **0.237** | 0.187 | 0.263 | 0.504 | 0.477 | 0.304 | 0.374 | 0.209 | 0.293 | 0.150 | 0.251 | 0.362 | 0.438 |
| **PEMS08** | | | | | | | | | | | | | | |
| 12 | 0.291(0.02) | 0.297(0.02) | 0.309 | 0.314 | 0.089 | 0.193 | 0.125 | 0.239 | 0.091 | 0.195 | 0.095 | 0.203 | 0.467 | 0.503 |
| 24 | 0.287(0.02) | 0.299(0.03) | 0.342 | 0.340 | 0.138 | 0.243 | 0.238 | 0.336 | 0.144 | 0.247 | 0.150 | 0.257 | 0.503 | 0.512 |
| 48 | 0.333(0.03) | 0.328(0.03) | 0.344 | 0.346 | 0.237 | 0.277 | 0.528 | 0.534 | 0.254 | 0.332 | 0.256 | 0.344 | 0.964 | 0.729 |
| 96 | 0.362(0.04) | 0.364(0.04) | 0.398 | 0.393 | 0.346 | 0.363 | 1.150 | 0.808 | 0.435 | 0.441 | 0.399 | 0.415 | 1.021 | 0.763 |
| **Avg** | 0.318 | 0.322 | 0.348 | 0.348 | 0.202 | 0.269 | 0.510 | 0.479 | 0.231 | 0.304 | 0.225 | 0.305 | 0.739 | 0.627 |

datasets (PEMS03, PEMS04, PEMS07, PEMS08) with lookback window $L = 96$ and prediction horizons $H \in \{12, 24, 48, 96\}$. Table 5 presents the results with MSE and MAE reported on the normalized scale.

Table 5 presents the results with MSE and MAE reported on the **normalized scale**. MSTN-Transformer and MSTN-BiLSTM show improved performance compared to baselines including iTransformer (Liu et al., 2024), PatchTST (Nie et al., 2023), and TSMixer (Chen et al., 2023). On PEMS03, MSTN-Transformer achieves competitive performance with average MSE of 0.156, outperforming iTransformer (0.458) by 65.9% and Autoformer (0.554) by 71.8%. On PEMS04, our model achieves an average MSE of 0.118, outperforming iTransformer (0.127) by 7.1% and Autoformer (0.764) by 84.6%. On PEMS07, MSTN-Transformer achieves an average MSE of 0.149, surpassing ModernTCN (0.304) by 51.0% and Autoformer (0.362) by 58.8%. On PEMS08, MSTN-Transformer achieves an average MSE of 0.318, outperforming Autoformer (0.739) by 57.0% and showing competitive performance compared to iTransformer (0.202). These results demonstrate that MSTN's multi-scale design effectively captures both short-term fluctuations and periodic patterns.

Figures 4 and 5 illustrate the 336-step forecasting performance of MSTN-Transformer, iTransformer, TSMixer, and Autoformer on PEMS03 and PEMS04 traffic datasets under the input-96-predict-336 setting. MSTN-Transformer (H=96) consistently achieves the lowest MSE across both datasets, with values of 0.182 on PEMS03 and 0.123 on PEMS04, significantly outperforming iTransformer (1.215 and 0.169), TSMixer (0.266 and 0.190), and Autoformer (0.710 and 0.853). MSTN-Transformer exhibits better forecasting behavior, showing smoother trajectories with fewer abrupt fluctuations compared to baseline models. Importantly, it consistently demonstrates better phase alignment with the ground truth, with predicted peaks and troughs occurring at nearly identical temporal locations, accurately preserving the periodic structure of traffic patterns. Such temporal fidelity is critical for practical traffic forecasting scenarios, where the correct timing of congestion events is often more important than exact amplitude matching.

Although MSTN-Transformer produces slightly reduced peak amplitudes in some cases, this conservative behavior reflects an error-minimizing bias of MSE-based optimization, favoring stable trend estimation over aggressive extrapolation. In the late horizon, MSTN-Transformer adapts to distribution shifts in the ground truth and tracks structural trends, whereas iTransformer and Autoformer exhibit more erratic variations and

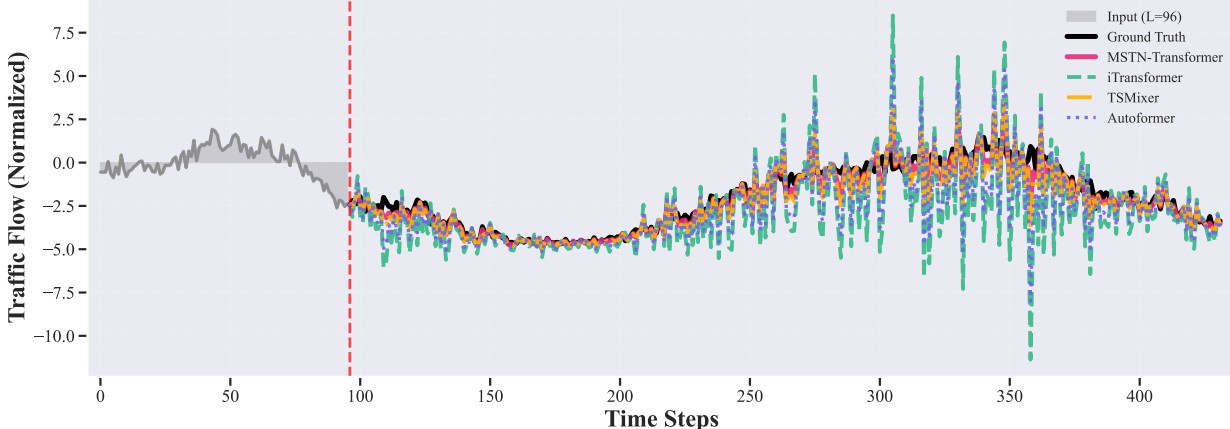

Figure 4: Visualization of PEMS03 dataset predictions by different models under the input-96-predict-336 setting.

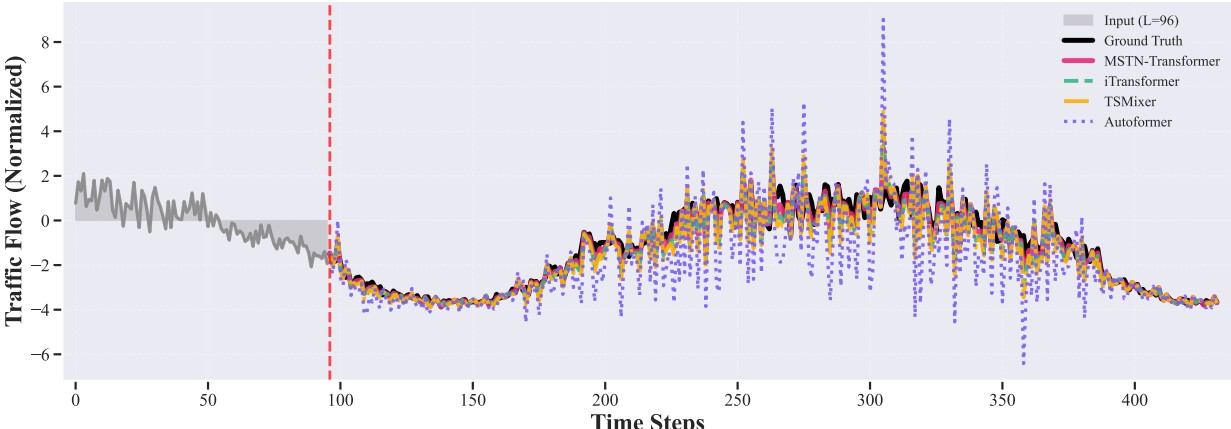

Figure 5: Visualization of PEMS04 dataset predictions by different models under the input-96-predict-336 setting.

larger deviations. These results demonstrate the effective long-horizon forecasting capability, robustness, and non-stationary adaptation ability of MSTN-Transformer. For clarity and readability, we visualize only representative and competitive baselines, while quantitative results for all compared models are reported in Table 5.

### 4.6 Time-Series Classification Results

Classification involves categorizing entire time series sequences. It is used to identify which type of event or pattern the sequence represents. For example, this could be the classification of a segment of sensor data as walking, running, falling, or collision activity, which is fundamental for automated monitoring and diagnosis systems. Performance is evaluated using the standard metric of classification accuracy.

Table 6a presents the classification results across 10 UEA standard benchmark datasets. The proposed MSTN framework is compared against SOTA time-series classification methods, including TimesNet (Wu et al., 2023), GPT-2(6) (Zhou et al., 2023), FEDformer (Zhou et al., 2022), and LightTS (Zhang et al., 2022). MSTN shows competitive performance across multiple datasets. MSTN-BiLSTM achieves improved accuracy on multiple datasets: EthanolConcentration (35.84%), Handwriting (59.88%), JapaneseVowels (99.41%), PEMS-SF (90.87%), and UWaveGestureLibrary (89.81%). MSTN-Transformer achieves better accuracy

Table 6: Comprehensive evaluation of MSTN across diverse time-series benchmarks. We evaluate each model five times and take the mean of the final results. **Red**/**Blue**: First/Second accuracy ranks. We evaluate each model five times and report the mean (standard deviation); results marked with underline are not significantly worse than the first rank (Wilcoxon signed-rank test with Holm–Bonferroni correction, $\alpha = 0.05$).

(a) Classification accuracy (%) comparison between MSTN (MSTN- Transformer and MSTN-BiLSTM) and SOTA baselines.

| Dataset | MSTN BiL.(ours) | MSTN Trans.(ours) | GPT2 (6) | Times-(Net) | Light TS | DLinear | Flow. | ETS. | FED. | Stat. | Auto. | Py. | In. | Re. | Trans. |
|---|---|---|---|---|---|---|---|---|---|---|---|---|---|---|---|
| Ethanol | **35.84 (3.18)** | 34.58 (3.71) | 31.9 | **35.7** | 29.7 | 32.6 | 33.8 | 28.1 | 31.2 | 32.7 | 31.6 | 30.8 | 31.6 | 31.9 | 32.7 |
| FaceDetect | 58.20 (1.26) | 57.26 (1.94) | 67.3 | **68.6** | 67.5 | 68.0 | 67.6 | 66.3 | 66.0 | 68.0 | **68.4** | 65.7 | 67.0 | **68.6** | 67.3 |
| Handwriting | **59.88 (2.41)** | **51.53 (3.34)** | 32.0 | 32.1 | 26.1 | 27.0 | 33.8 | 32.5 | 28.0 | 31.6 | 36.7 | 29.4 | 32.8 | 27.4 | 32.0 |
| Heartbeat | 80.39 (2.78) | **81.46 (4.55)** | 76.1 | 78.0 | 75.1 | 75.1 | 77.6 | 71.2 | 73.7 | 73.7 | 74.6 | 75.6 | **80.5** | 77.1 | 76.1 |
| JapaneseV | **99.41 (1.12)** | **99.21 (1.26)** | 98.6 | 98.4 | 96.2 | 96.2 | 98.9 | 95.9 | 98.4 | 99.2 | 96.2 | 98.4 | 98.9 | 97.8 | 98.7 |
| PEMS-SF | **90.87 (1.78)** | **89.92 (3.77)** | 82.1 | 89.6 | 88.4 | 75.1 | 83.8 | 86.0 | 80.9 | 87.3 | 82.7 | 83.2 | 81.5 | 82.7 | 82.1 |
| SCP1 | 86.32 (3.37) | 87.65 (3.47) | 93.2 | 91.8 | 89.8 | 87.3 | 92.5 | 89.6 | 88.7 | 89.4 | 84.0 | 88.1 | 90.1 | 90.4 | 92.2 |
| SCP2 | **58.86 (2.45)** | **60.37 (3.07)** | 59.4 | 57.2 | 51.1 | 50.5 | 56.1 | 55.0 | 54.4 | 57.2 | 50.6 | 53.3 | 53.3 | 56.7 | 53.9 |
| SpokenArabic | 98.54 (0.62) | 98.62 (0.59) | 99.2 | 99.0 | 100 | 81.4 | 98.8 | 100 | 100 | 100 | 100 | 99.6 | 100 | 97.0 | 98.4 |
| UWave | **89.81 (2.12)** | 77.89 (2.45) | 88.1 | 85.3 | 80.3 | 82.1 | 86.6 | 85.0 | 85.3 | 87.5 | 85.9 | 83.4 | 85.6 | 85.6 | 85.6 |
| Avg | **75.26** | 73.41 | 72.89 | **73.67** | 70.43 | 65.53 | 72.85 | 70.96 | 70.66 | 72.73 | 71.07 | 70.75 | 72.13 | 71.52 | 71.90 |

(b) Generalizability Study: Performance comparison of MSTN-BiLSTM and MSTN-Transformer with TimesNet, PatchTST, and prior specialized methods (RF, GB, DT, kNN, SAE, etc.) across seven international cross-domain datasets. **Cyan**/**Violet**: First/Second inference time ranks.

| Dataset | Det. | Dom. | Cnt. | A.Block | MSTN Acc.(%) | Size | I.Time | TimesNet Acc.(%) | I.Time | PatchTST Acc.(%) | I.Time | Prior Work Accuracy(%) |
|---|---|---|---|---|---|---|---|---|---|---|---|---|
| Rodegast (Rodegast et al., 2024a) | Sim. | Hum. | DE. | BiLSTM | **99.25** (0.20) | 1.34 | **1.59** | 99.18 | 5.07 | 98.64 | 4.80 | 91.00[RF,GB](Rodegast et al., 2024b) |
| | | | | Transf. | **99.39** (0.14) | 3.77 | **4.03** | | | | | |
| Boubezoul(Boubezoul et al., 2020) | Real. | Hum. | FR. | BiLSTM | **99.16** (0.25) | 1.03 | **1.41** | 93.00 | 4.20 | 91.37 | 4.68 | 91.59[DT](Elwy et al., 2023) |
| | | | | Transf. | **97.67** (0.19) | 3.54 | **1.70** | | | | | |
| UCI-HAR(Reyes-Ortiz et al., 2013) | Act. | Hum. | IT. | BiLSTM | **96.41** (0.22) | 1.68 | **2.22** | 91.38 | 45.30 | 93.21 | 4.36 | 83.35[Km,NB](Ismi et al., 2016) |
| | | | | Transf. | **96.84** (0.12) | 4.01 | **3.63** | | | | | |
| PAMAP2(Reiss, 2012) | Phy. | Hum. | US. | BiLSTM | **99.73** (0.09) | 1.64 | **0.31** | 95.13 | **0.46** | 98.02 | 3.21 | 90.00[kNN](Reiss & Stricker, 2012) |
| | | | | Transf. | **99.89** (0.07) | 4.13 | 1.90 | | | | | |
| ActBeC.(Dissanayake et al., 2025) | Calf. | Anim. | IE. | BiLSTM | **93.00** (0.26) | 3.95 | **1.76** | 90.65 | 8.53 | 62.45 | 2.20 | 84.00[RCCV](Dissanayake et al., 2025) |
| | | | | Transf. | **92.85** (0.20) | 4.14 | **0.29** | | | | | |
| MetroPT3(Davari et al., 2021a) | Met. | Mech. | PT. | BiLSTM | **93.33** (0.16) | 1.45 | **0.07** | 93.00 | **0.09** | 81.67 | 0.19 | 62.00[SAE](Davari et al., 2021b) |
| | | | | Transf. | **95.00** (0.13) | 3.98 | 0.69 | | | | | |
| NASA(Saxena & Goebel, 2008) | Eng. | Mech. | US. | BiLSTM | **11.68**[†] (0.28) | 1.12 | **2.25** | 15.56[†] | 4.52 | 31.59[†] | 2.31 | – |
| | | | | Transf. | **10.25**[†] (0.19) | 3.62 | **3.10** | | | | | |

**Abbreviations:** Det: Details (Sim: Simulator PTW data, Real: Real PTW Data, Act: Activity human, Phy: Physical activity, Calf: Calf Behavior, Met: Metro predictive maintenance, Eng: Engine sensor), Dom: Domain (Hum: Human Safety, Anim: Animal welfare, Mech: Mechanical), Cnt: Country (IN: India, FR: France, DE: Germany, IT: Italy, US: USA, IE: Ireland, PT: Portugal), A.Block: Architecture block (Transf. Transformer), I.Time: Inference time, †RMSE values.

on Heartbeat (81.46%) and ranks second on multiple datasets.MSTN-Transformer and MSTN-BiLSTM both achieve better performance (98.62% and 98.54%) on SpokenArabicDigits. In terms of overall average accuracy, MSTN-BiLSTM ranks first (75.26%) among all baselines compared, confirming the consistent improvement of the proposed MSTN framework.

## 4.7 Generalizability Study

A robust model should not be a specialist in just one area; it should adapt to new tasks and data distributions across different data and domains. This experiment evaluates the ability of the MSTN to transfer its knowledge across diverse domains, from human activity recognition to mechanical fault prediction. As summarized in Table 6b, the proposed MSTN framework demonstrates generalization across seven international benchmark datasets spanning human safety, activity recognition, animal welfare, and mechanical prognostics. Without domain-specific tuning, MSTN shows improved performance over state-of-the-art time series models—including TimesNet (Wu et al., 2023) and PatchTST (Nie et al., 2023)—as well as specialized prior methods (e.g., RF, GB, DT, kNN, SAE) reported in the Prior Work Accuracy column of Table 6b.

In human safety applications, MSTN-Transformer achieves 99.39% accuracy on German collision prediction data (Rodegast et al., 2024a), outperforming prior work (91.0%) and surpassing TimesNet (99.18%) and

Table 7: Ablation on time series imputation (Weather) with different masking ratios (12.5%, 25%, 37.5%, 50%). **Red**/**Blue**: First/Second ranks.

| MSTN Transformer Mask | Full MSE MAE | w/o CNN MSE MAE | w/o Transf. MSE MAE | w/o SE MSE MAE | w/o SDL MSE MAE | w/o SGF MSE MAE | w/o ETA MSE MAE |
|---|---|---|---|---|---|---|---|
| 12.5% | **0.007 0.014** | 0.009 0.018 | 0.014 0.023 | **0.008** 0.016 | **0.008 0.015** | 0.009 0.016 | **0.008 0.014** |
| 25% | **0.014 0.028** | 0.019 0.032 | 0.026 0.044 | 0.016 0.030 | **0.015** 0.030 | 0.016 0.031 | **0.015 0.029** |
| 37.5% | **0.021 0.042** | 0.024 0.046 | 0.038 0.065 | 0.022 0.044 | 0.023 0.044 | **0.021 0.043** | **0.020 0.042** |
| 50% | **0.029 0.058** | 0.033 0.062 | 0.053 0.092 | **0.030** 0.060 | **0.030 0.059** | 0.031 0.062 | 0.031 0.060 |

| MSTN BiLSTM | Full | | w/o CNN | | w/o BiLSTM | | w/o SE | | w/o SDL | | w/o SGF | | w/o ETA | |
|---|---|---|---|---|---|---|---|---|---|---|---|---|---|---|
| 12.5% | **0.009 0.018** | | 0.012 0.021 | | 0.013 0.023 | | **0.010 0.019** | | 0.011 **0.019** | | 0.011 0.020 | | **0.010 0.019** | |
| 25% | **0.018 0.035** | | 0.022 0.039 | | 0.026 0.045 | | **0.019 0.036** | | **0.019** 0.037 | | **0.019 0.036** | | **0.018** 0.037 | |
| 37.5% | **0.027** 0.053 | | 0.030 0.055 | | 0.038 0.066 | | 0.028 0.053 | | 0.028 **0.052** | | **0.027 0.052** | | **0.025 0.050** | |
| 50% | **0.036 0.070** | | 0.039 0.074 | | 0.053 0.092 | | **0.037 0.071** | | **0.037** 0.072 | | 0.038 0.072 | | 0.038 0.072 | |

PatchTST (98.64%). For activity recognition, it achieves 96.84% on UCI-HAR (Reyes-Ortiz et al., 2013), surpassing TimesNet (91.38%) and PatchTST (93.21%). On fall detection, MSTN-Transformer achieves 99.39% and MSTN-BiLSTM 99.16% on Boubezoul, outperforming TimesNet (93.00%) and PatchTST (91.37%). On PAMAP2, MSTN-Transformer achieves 99.89%, exceeding TimesNet (95.13%) and PatchTST (98.02%). For animal welfare, MSTN-BiLSTM achieves 93.00% on calf behavior recognition (Dissanayake et al., 2025), outperforming TimesNet (90.65%) and PatchTST (62.45%). In mechanical prognostics, MSTN-Transformer achieves 95.00% on MetroPT3, surpassing TimesNet (93.00%) and PatchTST (81.67%). On the NASA Turbofan dataset (Saxena & Goebel, 2008), MSTN-Transformer achieves 10.25 RMSE, outperforming TimesNet (15.56) and PatchTST (31.59). MSTN-BiLSTM also shows improved performance.

Beyond accuracy, Table 6b highlights the structural impact of our ETA mechanism. While traditional models maintain full sequence complexity throughout deep layers, MSTN restricts this heavy computation to the initial encoding stage. Consequently, MSTN-BiLSTM achieves inference times of 1.59 ms on Rodegast ($3.2\times$ faster than TimesNet) and 0.07 ms on MetroPT3 ($1.3\times$ faster than TimesNet's 0.09 ms). On PAMAP2, MSTN-BiLSTM reaches 0.31 ms, significantly outperforming PatchTST (3.21 ms) by $10.4\times$. This confirms that MSTN effectively breaks the traditional trade-off between modeling capacity and latency.

### 4.8 Ablation Study

An ablation study systematically removes individual components from the full model to evaluate their contributions. Table 7 presents the ablation results for imputation task on Weather dataset with varying mask ratios ({12.5%, 25%, 37.5%, 50%}). For MSTN-Transformer, six variants are tested: w/o CNN, w/o Transformer, w/o SE, w/o SGF, w/o SDL, and w/o ETA. For MSTN-BiLSTM, the variants include w/o CNN, w/o BiLSTM, w/o SE, w/o SGF, w/o SDL, and w/o ETA. For MSTN-Transformer, the full model achieves improved performance, with MSE of 0.007, 0.014, 0.021, and 0.029 across increasing mask ratios. The Transformer branch is most critical, with its removal causing the largest degradation (e.g., MSE increases from 0.007 to 0.014 at 12.5% mask, and from 0.029 to 0.053 at 50% mask). The w/o CNN variant also shows reasonable performance, while the w/o SE and w/o ETA variants demonstrate competitive performance, often ranking second. For MSTN-BiLSTM, the BiLSTM branch proves most important, with its removal causing substantial performance drops (e.g., MSE rises from 0.009 to 0.013 at 12.5% mask, and from 0.036 to 0.053 at 50% mask). The w/o ETA variant achieves the best performance at 37.5% mask (0.025 MSE), outperforming the full model (0.027 MSE), suggesting that preserving temporal information is particularly beneficial for challenging imputation scenarios. Notably, removing ETA (w/o ETA) yields mixed results: for MSTN-Transformer, it shows competitive performance across all masks (0.008, 0.015, 0.020, 0.031 MSE); for MSTN-BiLSTM, w/o ETA achieves better performance at 37.5% mask (0.025 MSE, outperforming full model's 0.027), indicating that ETA is often beneficial for both architectures.

**Progressive Bottom-Up Validation:** To further verify that every component is necessary, we evaluate a progressive construction of MSTN-BiLSTM on the Weather imputation task (12.5% mask) using results from Table 7. The CNN-only baseline (w/o BiLSTM) achieves 0.013 MSE. Adding the BiLSTM branch reduces MSE to 0.010. Incorporating SE and SDL yields 0.011 MSE. Adding SGF improves to 0.010 MSE. Finally,

Table 8: Ablation studies of MSTN-Transformer and MSTN-BiLSTM on PEMS04 Dataset. **Red**/**Blue**: First/Second ranks (based on MSE).

| MSTN Transformer H | Full MSE MAE | w/o CNN MSE MAE | w/o Transf. MSE MAE | w/o SE MSE MAE | w/o SDL MSE MAE | w/o SGF MSE MAE | w/o ETA MSE MAE |
|---|---|---|---|---|---|---|---|
| 12 | **0.105 0.210** | 0.110 0.216 | 0.125 0.241 | 0.108 0.214 | 0.108 **0.213** | **0.107** 0.214 | **0.107 0.213** |
| 24 | **0.111** 0.217 | 0.118 0.230 | 0.131 0.246 | 0.116 0.224 | 0.116 0.223 | **0.111 0.216** | **0.110 0.215** |
| 48 | **0.119 0.229** | 0.125 0.240 | 0.136 0.252 | **0.120 0.231** | 0.122 0.233 | **0.120** 0.232 | **0.120** 0.232 |
| 96 | **0.126 0.235** | 0.134 0.247 | 0.154 0.273 | **0.127 0.237** | **0.127** 0.240 | 0.128 0.245 | 0.128 0.241 |

| MSTN BiLSTM | Full | w/o CNN | w/o BiLSTM | w/o SE | w/o SDL | w/o SGF | w/o ETA |
|---|---|---|---|---|---|---|---|
| 12 | **0.114 0.222** | 0.122 0.233 | 0.126 0.241 | 0.116 0.228 | **0.115** 0.230 | 0.116 0.227 | **0.115 0.225** |
| 24 | **0.120 0.228** | 0.127 0.244 | 0.132 0.248 | 0.122 0.233 | **0.120** 0.233 | 0.123 0.232 | **0.119 0.230** |
| 48 | **0.127** 0.239 | 0.134 0.248 | 0.144 0.265 | **0.128 0.241** | 0.130 0.247 | **0.128 0.241** | **0.128 0.238** |
| 96 | **0.137 0.247** | 0.144 0.257 | 0.148 0.269 | 0.140 0.250 | **0.139 0.248** | **0.139** 0.249 | 0.140 0.249 |

Table 9: Ablation studies of MSTN-Transformer and MSTN-BiLSTM across different tasks. (a) Classification on 10 UEA, (b) Generalizability Study. **Red**/**Blue**: First/Second ranks.

(a) Ablation study on classification benchmarks.

| MSTN Transformer | Full | w/o CNN | w/o Transf. | w/o SE | w/o SDL | w/o SGF | w/o ETA |
|---|---|---|---|---|---|---|---|
| PEMS-SF | **89.92** | 86.25 | 85.11 | 89.26 | 85.44 | **89.36** | 81.82 |
| JapaneseVowels | **99.21** | 97.23 | 98.14 | 98.47 | 98.32 | 98.41 | **98.44** |
| SpokenArabic | **98.62** | 98.00 | 98.28 | **98.46** | 98.33 | 98.42 | 98.17 |

| MSTN BiLSTM | Full | w/o CNN | w/o BiLSTM | w/o SE | w/o SDL | w/o SGF | w/o ETA |
|---|---|---|---|---|---|---|---|
| PEMS-SF | **90.87** | 88.60 | 87.46 | 90.40 | 87.46 | 90.65 | **90.91** |
| JapaneseVowels | **99.41** | 96.97 | 99.05 | 99.14 | **99.17** | 99.00 | 98.44 |
| SpokenArabic | **98.54** | 97.86 | 98.14 | **98.29** | 98.23 | 98.19 | 98.06 |

(b) Ablation study on generalizability datasets.

| MSTN Transformer | Full | w/o CNN | w/o Transf. | w/o SE | w/o SDL | w/o SGF | w/o ETA |
|---|---|---|---|---|---|---|---|
| PTW Simulation | **99.39** | 98.89 | 98.73 | 99.04 | 99.08 | 98.93 | **99.25** |
| PAMAP2 | **99.89** | 99.18 | 99.34 | 99.23 | **99.65** | 99.39 | 99.59 |

| MSTN BiLSTM | Full | w/o CNN | w/o BiLSTM | w/o SE | w/o SDL | w/o SGF | w/o ETA |
|---|---|---|---|---|---|---|---|
| PTW Simulation | **99.25** | 98.89 | 98.56 | **99.11** | 99.05 | 99.10 | 99.08 |
| PAMAP2 | **99.73** | 99.39 | 99.35 | 99.62 | **99.68** | 99.56 | 99.51 |

integrating ETA achieves the best result of 0.009 MSE. Each addition consistently improves or maintains performance, confirming that the full architecture is not over-engineered and that all components contribute synergistically.

Table 8 presents ablation studies for MSTN-Transformer and MSTN-BiLSTM on PEMS04 dataset across four horizons (H=12, 24, 48, 96). For MSTN-Transformer, the full model ranks first at H=12 (0.105/0.210), H=48 (0.119/0.229), and H=96 (0.126/0.235). w/o ETA achieves best at H=24 (0.110/0.215) and shows competitive performance elsewhere. w/o Transformer yields worst performance across all horizons (MSE up to 0.154), confirming the critical role of the Transformer branch. For MSTN-BiLSTM, the full model ranks first at H=12 (0.114/0.229), H=48 (0.127/0.246), and H=96 (0.137/0.247). w/o ETA achieves best at H=24 (0.119/0.239), outperforming the full model. w/o BiLSTM performs worst (MSE up to 0.148), confirming the essential role of bidirectional sequential modeling. Overall, the full architecture provides improved performance, while removing ETA benefits specific horizons (H=24), suggesting temporal pooling configuration can be task-dependent. The full models also show better performance while preserving a balance between predictive accuracy and architectural complexity. These findings indicate that removing ETA and SDL are effective design choices that can be adapted based on specific dataset or tasks, whereas the full model with ETA offers a more balanced trade-off between performance and model complexity.

On classification benchmarks (PEMS-SF, JapaneseVowels, SpokenArabic) (Table 9a), the full MSTN-Transformer achieves 89.92%, 99.21%, and 98.62% accuracy respectively, while MSTN-BiLSTM achieves

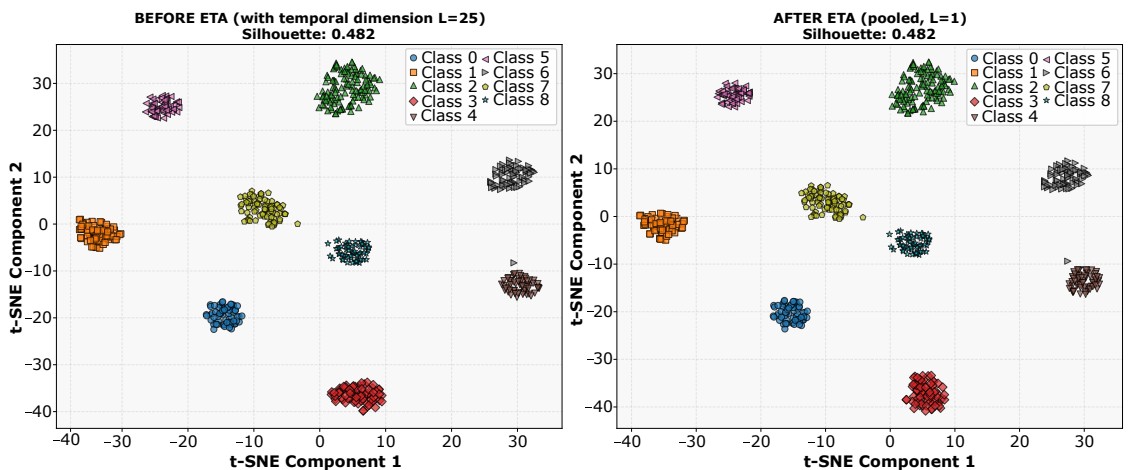

Figure 6: t-SNE visualization: Effect of ETA on JapaneseVowels dataset ($L = 25 \to 1$, 9 classes). Despite $25\times$ temporal compression, the silhouette score remains unchanged (0.482 before vs. 0.482 after).

90.87%, 99.41%, and 98.54% accuracy respectively. Ablation confirms that both branches are essential, with w/o ETA causing drops in most cases (e.g., MSTN-Transformer on PEMS-SF falls from 89.92% to 81.82%, MSTN-BiLSTM on JapaneseVowels drops from 99.41% to 98.44%), suggesting the importance of ETA in most cases. Notably, w/o ETA surprisingly outperforms the full model for MSTN-BiLSTM on PEMS-SF (90.91% vs 90.87%) and achieves competitive results, indicating that ETA's benefit varies across datasets and architectures. The SE and SDL modules provide complementary gains.

On generalizability studies (PTW Simulation, PAMAP2) (Table 9b), both architectures show better performance, with MSTN-Transformer achieving 99.39%/99.89% and MSTN-BiLSTM achieving 99.25%/99.73% accuracy. The Transformer/BiLSTM branches are most critical, while w/o ETA achieves competitive results, (e.g., MSTN-Transformer on PTW Simulation drops from 99.39% to 99.25%; MSTN-BiLSTM on PAMAP2 drops from 99.73% to 99.51%), showing ETA's importance for generalization. The SE and SDL modules provide consistent improvements across both architectures and datasets. In summary, the ablation study validates that each architectural component is essential. The Transformer/BiLSTM branch, convolutional branch, SE module, SDL, and ETA operate synergistically, with the full integrated model delivering the best results across diverse temporal datasets.

## 4.9 Comparing Representation Collapse: Models Maintaining L vs. ETA with t-SNE

To empirically verify that ETA preserves discriminative information despite collapsing the temporal dimension. We perform feature-space visualization using t-distributed Stochastic Neighbor Embedding (t-SNE) visualization (Van der Maaten & Hinton, 2008) on features extracted before and after the ETA (pooling) operation on two diverse datasets: JapaneseVowels (multi-class, $L = 25$) and Heartbeat (binary, $L = 405$). We quantify cluster quality using the silhouette score (Rousseeuw, 1987).

Figure 6 shows the visualization of t-SNE on the JapaneseVowels dataset (9 classes, 640 samples, L=25). The silhouette score remains identical before and after ETA (0.482), demonstrating that $25\times$ temporal compression preserves all class-discriminative information. Figure 7 shows the visualization of t-SNE on the Heartbeat dataset (binary classification, 409 samples, L=405). Remarkably, even at $405\times$ compression, the silhouette score remains identical (0.110 before vs. 0.110 after), confirming that ETA preserves discriminative information even at extreme compression ratios.

These results indicate that ETA preserves discriminative information across diverse datasets, with identical silhouette scores before and after ETA (pooling). Models that maintain the full temporal dimension achieve the same feature quality as ETA's compressed representation ($L = 1$), indicating that there is little advantage in keeping $L$ throughout the network. Furthermore, ETA achieves effective temporal compression even at

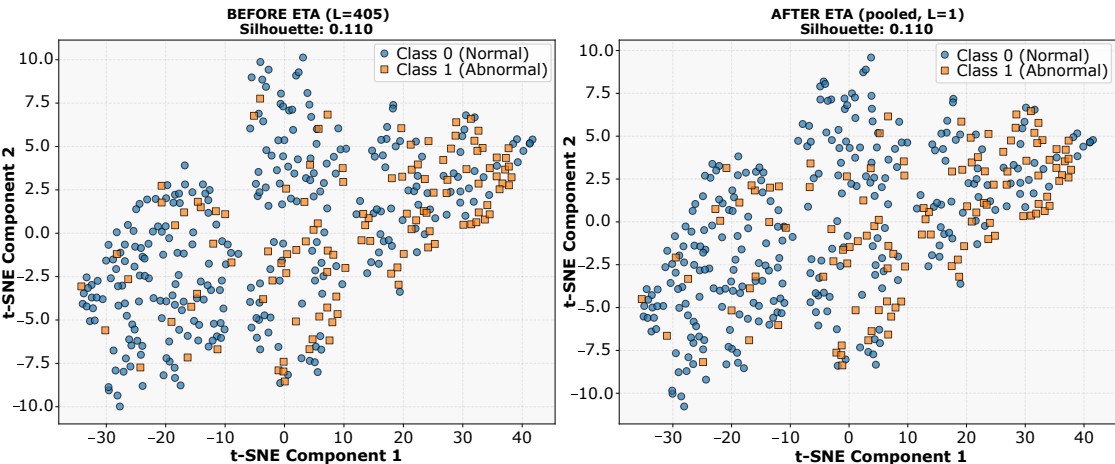

Figure 7: t-SNE visualization: Effect of ETA on Heartbeat dataset ($L = 405 \rightarrow 1$, binary). Despite $405\times$ temporal compression, the silhouette score remains unchanged (0.110 before vs. 0.110 after).

extreme ratios ($405\times$), enabling $\mathcal{O}(1)$ complexity for downstream modules after the initial encoder without sacrificing representation quality.

### 4.10 Model Complexity and Edge-AI Deployability

#### 4.10.1 Complexity Analysis

In this analysis, we denote the length of the lookback window as $L$, the number of variate features (channels) as $C$, and the task-specific output dimension as $K$.

The MSTN framework addresses the fundamental trade-off in temporal modeling between representational capacity and computational efficiency. Although the proposed variants exhibit distinct encoding complexities, MSTN-BiLSTM maintains strict linearity with respect to sequence length and MSTN-Transformer incurring a quadratic cost. They both converge on a unified efficiency strategy: ETA. This is achieved because both the parallel CNN and sequence modeling branches utilize immediate temporal pooling to eliminate sequence-length dependence in subsequent layers. Specifically, both the parallel CNN and sequence modeling branches (Transformer/BiLSTM) independently condense their temporal features via Global Average Pooling and Sequence Mean Pooling, respectively (reducing the sequence dimension from $L$ to 1). Following concatenation, the subsequent SGF, SE-Block, and SDL refinement stages operate on static feature vectors. This strategic design ensures that the refinement pipeline scales as $\mathcal{O}(C^2)$ with respect to channels but remains constant-time ($\mathcal{O}(1)$) after ETA with respect to the input length $L$, effectively isolating the heavy temporal computation ($\mathcal{O}(L)$ for BiLSTM or $\mathcal{O}(L^2)$ for Transformer) to the initial encoding stage.

**Component-Wise Analysis of MSTN-Transformer:** The complexity is dominated by the sequence encoding stage. First, the Multi-Scale CNN extracts local features with linear complexity $\mathcal{O}(CL)$. Second, the Transformer Encoder applies self-attention; as this requires computing an $L \times L$ attention matrix, it introduces the dominant quadratic term $\mathcal{O}(CL^2)$. Following this, Temporal Pooling aggregates the sequence with linear cost $\mathcal{O}(CL)$. Crucially, the Refinement stage operates on the pooled vector, scaling with channel interactions as $\mathcal{O}(C^2)$. Finally, the Projection Head maps features to the output with complexity $\mathcal{O}(CK)$. Summing these components, the total asymptotic complexity is $\mathcal{O}(CL^2+C^2+CK)$, which scales quadratically with sequence length $L$.

**Component-Wise Analysis of MSTN-BiLSTM:** In contrast, the MSTN-BiLSTM variant achieves strict linear efficiency with respect to sequence length. The BiLSTM Encoder processes the sequence recurrently without generating a global attention map, resulting in a linear complexity of $\mathcal{O}(CL)$. Similar to the Transformer variant, Temporal Pooling ($\mathcal{O}(CL)$) condenses the time dimension. The Refinement stage contributes

Table 10: Comprehensive evaluation of MSTN. Inference time (IT, in ms) of MSTN variants compared to TimesNet and PatchTST across seven international datasets. **Red**/**Blue**: First/Second ranks.

| Dataset | MSTN | Size (MB) | IT (ms) | TimesNet IT | PatchTST IT |
|---|---|---|---|---|---|
| Rodegast (Rodegast et al., 2024a) | MSTN-Trans. | 3.77 | **4.03** | 5.07 | 4.80 |
| | MSTN-BiLSTM | 1.34 | **1.59** | | |
| Boubezoul (Boubezoul et al., 2020) | MSTN-Trans. | 3.54 | **1.70** | 4.20 | 4.68 |
| | MSTN-BiLSTM | 1.03 | **1.41** | | |
| UCI-HAR (Reyes-Ortiz et al., 2013) | MSTN-Trans. | 4.01 | **3.63** | 45.30 | 4.36 |
| | MSTN-BiLSTM | 1.68 | **2.22** | | |
| PAMAP2 (Reiss, 2012) | MSTN-Trans. | 4.13 | 1.90 | **0.46** | 3.21 |
| | MSTN-BiLSTM | 1.64 | **0.31** | | |
| ActBeCalf (Dissanayake et al., 2025) | MSTN-Trans. | 4.14 | **0.29** | 8.53 | 2.20 |
| | MSTN-BiLSTM | 3.95 | **1.76** | | |
| MetroPT3 (Davari et al., 2021a) | MSTN-Trans. | 3.98 | 0.69 | **0.09** | 0.19 |
| | MSTN-BiLSTM | 1.45 | **0.07** | | |
| NASA (Saxena & Goebel, 2008) | MSTN-Trans. | 3.62 | **3.10** | 4.52 | 2.31 |
| | MSTN-BiLSTM | 1.12 | **2.25** | | |

$\mathcal{O}(C^2)$, and the Projection Head contributes $\mathcal{O}(CK)$. Since the encoder avoids quadratic operations in $L$, the total start-to-end complexity remains highly efficient at $\mathcal{O}(CL + C^2 + CK)$, which scales linearly with $L$.

This theoretical efficiency is empirically validated by the structural footprint of the model. As verified through our implementation, the MSTN-Transformer variant maintains a fixed core of $\sim$1.06 million parameters, while the MSTN-BiLSTM variant is highly optimized with only $\sim$0.40 million parameters. By ensuring a task-agnostic backbone, MSTN maintains consistent complexity across diverse datasets, meeting the requirement for architectural invariance. The transition from $\mathcal{O}(L)$ or $\mathcal{O}(L^2)$ in the encoding stage to $\mathcal{O}(1)$ in the refinement stage is the main reason for this low parameter count; the front-end encoder itself retains its original complexity ($\mathcal{O}(L^2)$ for Transformer, $\mathcal{O}(L)$ for BiLSTM), allowing the model to perform effective feature recalibration without an additive parameter penalty.

In our implementation with lookback window $L = 96$, the front-end Transformer encoder (a single encoder block) computes self-attention over 96 time steps. This results in $96^2 = 9,216$ attention weight computations per head per layer. Without ETA, the subsequent fusion, SE, SDL, and prediction modules would also need to process the full sequence length of 96, compounding the computational cost. With ETA, these downstream modules operate on a single aggregated vector ($L = 1$), reducing their complexity from $\mathcal{O}(L)$ or $\mathcal{O}(L^2)$ to $\mathcal{O}(1)$. For $L = 96$, this eliminates approximately $96\times$ to $9,216\times$ operations in the refinement stages, depending on the module. This practical reduction complements the theoretical complexity analysis and directly contributes to the sub-millisecond inference latency.

### 4.10.2 Comparative Analysis and Edge-AI Deployability

Recent efficient architectures demonstrate varying complexity profiles: linear models such as DLinear (Zeng et al., 2022) achieve $\mathcal{O}(CL)$ complexity, while spectral approaches like FEDformer (Zhou et al., 2022) and TimesNet (Wu et al., 2023) achieve $\mathcal{O}(CL \log L)$ complexity. iTransformer achieves $\mathcal{O}(C^2 + CL + CH)$ complexity by treating each variate as a token, PatchTST achieves $\mathcal{O}(CL^2 + CH)$ via patch-based tokenization, and Standard Transformers ($\mathcal{O}(N \cdot CL^2)$, where $N$ is the number of transformer layers) incur heavier costs.

Against this landscape, the proposed MSTN-BiLSTM achieves strict linear temporal complexity $\mathcal{O}(CL + C^2 + CK)$, which is optimal for the predominant real-world scenario where the sequence length exceeds the channel dimension ($L \gg C$). Meanwhile, MSTN-Transformer scaling as $\mathcal{O}(CL^2 + C^2 + CK)$ optimizes practical runtime by confining the quadratic term to a single encoder layer rather than a deep stack ($N$ layers) as seen in iTransformer or standard Transformer architectures. This strategic design allows MSTN to balance the high-capacity modeling of Transformers with the latency requirements of Edge-AI, delivering SOTA performance with minimized computational overhead.

One thing to note is that despite the MSTN-BiLSTM's better theoretical scaling ($\mathcal{O}(L)$) than the MSTN-Transformer's $\mathcal{O}(L^2)$, the latter runs faster than the former. This phenomenon is due to the fundamental conflict between the model's structure and the way modern acceleration architectures process the data. The

BiLSTM encoder is inherently recurrent, meaning that the calculation of the hidden state $h_t$ is strictly dependent on the previous state $h_{t-1}$. This sequential data dependency prevents the hardware's parallel processing units from being fully utilized, forcing the computation into a slow, iterative loop (low hardware utilization). In contrast, the Transformer executes its $\mathcal{O}(L^2)$ self-attention via highly optimized (Vaswani et al., 2023), complete matrix parallelization. The core operation, the $QK^T$ matrix multiplication, is performed simultaneously across thousands of processing elements. This high utilization of the hardware's parallel capacity, confined to a single encoder layer before pooling, ensures the Transformer's execution speed drastically outweighs the theoretical linearity of the BiLSTM's sequential cost, resulting in lower practical latency across high-throughput platforms.

Standard Transformers maintain the full sequence length $L$ across a multi-layer stack of $N$ blocks for self-attention operations, resulting in cumulative attention cost proportional to $N \cdot \mathcal{O}(CL^2)$. In contrast, MSTN applies attention in its encoder, then immediately pools $(L \to 1)$. Standard Transformers maintain $L$ across all layers, while MSTN eliminates $L$-dependence after encoding. Consequently, the downstream modules after ETA operate on static feature vectors with $\mathcal{O}(1)$ complexity relative to $L$ for operations involving sequence length, while the encoder retains its $\mathcal{O}(CL)$ (BiLSTM) or $\mathcal{O}(CL^2)$ (Transformer) complexity. By eliminating the recursive application of attention across deep layers, MSTN reduces the sequence-dependent FLOPs. While MSTN-BiLSTM avoids quadratic attention costs entirely by employing BiLSTM encoding with $\mathcal{O}(CL)$ complexity. When combined with the remaining pipeline components (CNN, refinement, and projection), the total complexity is $\mathcal{O}(CL + C^2 + CK)$, which scales linearly with sequence length $L$.

The inference analysis in Table 10 shows that MSTN maintains consistently low latency across seven heterogeneous datasets while maintaining a small footprint (1.03-4.14 MB). On the Rodegast dataset, MSTN-BiLSTM achieves 1.59 ms inference time, which is 3.2× faster than TimesNet (5.07 ms) and 3.0× faster than PatchTST (4.80 ms). On the Boubezoul dataset, MSTN-BiLSTM achieves 1.41 ms, outperforming TimesNet (4.20 ms) by 3.0× and PatchTST (4.68 ms) by 3.3×. On the ActBeCalf dataset, MSTN-Transformer achieves the fastest inference time of 0.29 ms, outperforming TimesNet (8.53 ms) by 29.4× and PatchTST (2.20 ms) by 7.6×. On the MetroPT3 dataset, MSTN-BiLSTM achieves an exceptionally low 0.07 ms, surpassing TimesNet (0.09 ms) and PatchTST (0.19 ms). These findings highlight the scalability and deployment efficiency of MSTN across both low-channel and high-dimensional sensor datasets, underscoring its suitability for latency-sensitive and edge-oriented time-series applications. Finally, in terms of comparative complexity, MSTN shows an advantage over the current SOTA frameworks. The proposed MSTN-BiLSTM and MSTN-Transformer variants utilize fixed cores of only ∼0.40 million and ∼1.06 million parameters.

Inference times reported in Table 10 were measured on an NVIDIA A100 GPU with a batch size of 32. We performed 10 warm-up iterations before measurement, followed by 5 independent forward passes whose results were averaged. Data loading and preprocessing were excluded from timing; only the forward pass through the model was measured. GPU-only results are reported as our deployment target is GPU-accelerated edge devices. The same protocol was applied to all compared models (TimesNet, PatchTST) to ensure fair comparison. This small memory footprint, combined with sub-millisecond inference latency, leads to better deployability on Edge-AI devices.

## 4.11 Limitations and Failure Cases

Although MSTN achieves state-of-the-art or competitive performance on most benchmarks, we analyze cases where specialized architectures outperform it, revealing important design trade-offs and current limitations. MSTN shows limitations on two imputation datasets (ETTh2, ETTm2), one forecasting dataset (PEMS08), and three classification datasets (FaceDetection, SelfRegulationSCP1, SpokenArabicDigits) where specialized architectures achieve better performance.

For the Imputation task, on ETTh2 and ETTm2, GPT2(3) achieves better results (0.048 MSE on ETTh2; 0.021 MSE on ETTm2) compared to MSTN-Transformer (0.247 MSE on ETTh2; 0.151 MSE on ETTm2). The autoregressive design of GPT2(3) captures high-frequency patterns more effectively than MSTN's multi-scale design on these datasets. However, on ETTh1, ETTm1, ECL, and Weather, MSTN achieves improved performance, particularly on ECL where MSTN-BiLSTM ranks first (0.071 MSE). For the Forecasting task, on PEMS08, iTransformer achieves better average MSE (0.202) compared to MSTN-Transformer (0.318)

and MSTN-BiLSTM (0.348). The inverted transformer design of iTransformer, which embeds each variate independently into a token, better handles high-dimensional (170 sensors) traffic data with complex spatial correlations.

For the Classification task, TimesNet's 2D modeling exceeds MSTN on FaceDetection with 68.6% accuracy (MSTN-BiLSTM: 58.20%, MSTN-Transformer: 57.26%), as it better captures the periodic patterns in facial data by transforming 1D sequences into 2D representations. On SelfRegulationSCP1, the pre-trained GPT2(6) model achieves 93.2% accuracy, outperforming MSTN-Transformer (87.65%) and MSTN-BiLSTM (86.32%), benefiting from its exposure to diverse sequential patterns during pre-training. On SpokenArabicDigits, LightTS achieves perfect 100% accuracy, while MSTN-Transformer achieves 98.62% and MSTN-BiLSTM achieves 98.54%, demonstrating that simpler architectures can be surprisingly effective on certain voice recognition tasks.

These gaps reflect the fundamental trade-off in MSTN's design: prioritizing general-purpose multi-scale learning over specialized inductive biases. By focusing on broad applicability across heterogeneous scenarios, MSTN achieves better performance on most benchmarks but may be outperformed on datasets where explicitly optimized architectures have advantages. This trade-off reflects our design philosophy, positioning MSTN as a universal temporal model for diverse real-world applications rather than one tailored to particular datasets or domain-specific tasks. The modular architecture of MSTN allows practitioners to adapt it to their needs—for instance, by replacing the pooling-based ETA with a learnable temporal aggregation mechanism for high-frequency data, or by incorporating domain-specific preprocessing. These extensions are promising directions for future work.

### 4.12 Representational Limits of Linear Decoder

The linear decoder $\hat{Y}_{1:H} = W_f \mathbf{z}_{\text{final}} + b_f$ maps a 192-dimensional latent vector to the forecast horizon $H$. For $H = 96$, the latent vector must represent $96 \times F$ output features, where $F$ is the number of sensors, which presents a compression challenge. However, three factors explain its empirical success in reconstructing complex traffic patterns including rush-hour peaks and off-peak periods:

(1) $\mathbf{z}_{\text{final}}$ is information-dense, capturing multi-scale features via CNN (local patterns), global dependencies via BiLSTM/Transformer, and adaptive fusion via SGF, SE, and SDL. This single vector encodes the essential temporal characteristics of the entire input sequence across all sensors simultaneously.

(2) $W_f$ learns 192 basis temporal patterns spanning different frequencies and phase shifts. The forecast is a linear combination $\hat{Y} = \sum_{i=1}^{192} \alpha_i \phi_i(t)$, where each basis $\phi_i(t)$ can represent different temporal dynamics—from low-frequency trends to high-frequency fluctuations—and coefficients $\alpha_i$ are derived from $\mathbf{z}_{\text{final}}$. This is analogous to learning a Fourier-like basis, but adapted to the data distribution.

(3) Empirical Performance: On PEMS03, PEMS04, and PEMS07, MSTN-Transformer achieves average MSE of 0.156, 0.115, and 0.149 respectively, ranking first among all baselines (Table 5). The linear decoder effectively reconstructs complex traffic patterns including rush-hour peaks and off-peak periods. Ablation studies (Table 8) confirm that all components contribute at $H = 96$, with removal of the Transformer branch increasing MSE by up to 22%.

For longer forecasting horizons or datasets with higher dimensionality, the compression ratio may exceed the representational capacity of 192 bases; we recommend hierarchical decoding— a direction for future work.

## 5 Conclusions

This work introduces MSTN, a DL framework founded on a hierarchical multi-scale modeling principle and defined by its strategic use of ETA. MSTN integrates core components: a multi-scale convolutional encoder that constructs a hierarchical feature pyramid; a sequential modeling component for long-term global dependency modeling, empirically validated with BiLSTM and Transformer variants; and a SGF mechanism augmented with SE and SDL to enable dynamic, context-aware feature integration. Existing methods often rely on fixed-scale structural priors (e.g., patch-based tokenization, fixed frequency transformations, or frozen backbone architectures). These limitations have historically forced a compromise between scale diversity and

inference speed. In contrast, our approach efficiently integrates convolutional local pattern extraction with long-term global dependency modeling, leveraging ETA via an adaptive SGF mechanism. This integrated design enables the model to capture multi-scale temporal features while maintaining computational efficiency, a combination we demonstrate yields SOTA performance.

Across extensive evaluations, MSTN demonstrates SOTA performance on 21 of 27 benchmark datasets spanning imputation, long-term forecasting, classification, and cross-dataset generalization. Notably, this improved performance is achieved with exceptional parameter efficiency. MSTN achieves this competitive performance with substantially fewer parameters than comparable SOTA methods: ~0.40M (MSTN-BiLSTM) and ~1.06M (MSTN-Transformer) parameters. MSTN-Transformer achieves improved imputation performance on the Weather dataset, with an average MSE of 0.018, outperforming TimesNet (0.030) by 40.0% and PatchTST (0.033) by 45.5%, all within a compact footprint, while MSTN-BiLSTM shows competitive performance. This dual advancement in both performance and efficiency is a distinguishing feature from prior work. This outcome, particularly the Transformer's better speed despite its quadratic complexity, validates the MSTN's ETA strategy, showing that maximizing hardware parallelization in the encoding stage is often more critical for latency than maintaining theoretical linear complexity. The model also exhibits better cross-domain generalization, with its small model size (1.03–4.14 MB) making it suitable for edge deployment. Ablation studies confirm that each component is critical, with the Transformer/BiLSTM sequence modeling core providing a foundational capability and the multi-scale and fusion modules delivering significant competitive gains.

Future work will explore cross-variate attention mechanisms and large-scale pre-training to further enhance the capabilities of MSTN as a foundation model for time series understanding. Our work paves the way for efficient multi-scale temporal modeling under resource constraints, enabling real-time applications across diverse domains from healthcare to industrial monitoring.

## Data and Code Availability

All datasets used in this study are publicly available from their respective sources as cited throughout the paper. The source code for MSTN is available at: `https://github.com/SumitPTW/MSTN`.

## Competing Interests

The authors declare no conflict of interest.

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
