# OpenReview forum: "MSTN: A Lightweight and Fast Model for General Time-Series Analysis"
_TMLR — Accepted by TMLR_

### Review · Reviewer_EDhk · 2026-03-09

**Summary Of Contributions:**

This paper proposes MSTN, a lightweight multi-scale architecture for general multivariate time-series analysis. The model combines a CNN branch for local pattern extraction with a Transformer or BiLSTM branch for sequence modeling, and introduces Early Temporal Aggregation to compress temporal features into compact representations before the final fusion and prediction stages. Built on this design, MSTN is applied to imputation, long-term forecasting, classification, and cross-domain generalization tasks. Experiments on a broad set of benchmarks show that the method achieves strong overall performance while maintaining a relatively small model size and favorable inference efficiency.

**Additional Comments:**

See Requested Changes

**Audience:**

Yes

**Audience Explanation:**

May be of interest to a subset of TMLR readers, particularly those working on time-series methods and applications.

**Claims And Evidence:**

No

**Claims Explanation:**

The submission does not provide sufficiently accurate, convincing, and clear evidence to fully support its claims. Although the paper proposes a framework that integrates multiple components (e.g., CNN branches, Transformer/BiLSTM branches, and modules such as SGF, SE, MHA, and ETA), the design motivation and the specific role of each module are not clearly explained, making it difficult to understand the necessity and advantages of the overall architecture. In addition, the definition of the loss in the interpolation task remains somewhat ambiguous, which leads to uncertainty regarding the supervision objectives during training and evaluation. The experimental evidence is also limited, as the main results are reported without statistical information such as standard deviations or significance tests, making it hard to assess the reliability of the improvements. Furthermore, the reporting of evaluation metrics is inconsistent across sections, and some datasets lack sufficient baseline comparisons, which weakens the strength of the empirical claims. Finally, the paper provides limited analysis of failure cases and does not sufficiently investigate why the proposed method underperforms on several datasets.

**Requested Changes:**

1. The methodology section's explanation of the design motivation is still insufficient. While the paper combines CNN branches, Transformer/BiLSTM branches, and modules such as SGF, SE, MHA, and ETA, the current main body focuses primarily on introducing the model's components and their connections. The explanation of why this overall architecture was adopted, the problems each module addresses, and the advantages of this combination compared to simpler alternatives is not clear enough. Therefore, there is room for further strengthening of the core innovations of the methodology and the necessity of each module's setup.
2. The definition of loss in the interpolation task remains somewhat ambiguous. The paper points out that the masked MSE in Eq. (10) is only calculated on the observations, but the experimental setup shows that some time points are randomly masked in the MCAR scenario for interpolation, which makes the supervision objectives in the training and evaluation phases unclear.
3. The statistical support for the experimental conclusions remains insufficient. The main results table primarily reports the MSE, MAE, or accuracy of a single experiment, but lacks statistical information such as standard deviation, confidence intervals, or significance tests. Therefore, it is currently difficult to determine whether the performance improvement is stable and statistically reliable.
4. The reporting of metrics in the classification experiments is inconsistent. Section 3.4.2 explicitly states that classification tasks will report Accuracy, Precision, Recall, and F1-Score, but the main classification table only provides Accuracy; these metrics primarily appear in the cross-domain generalization table, making the experimental organization somewhat inconsistent.
5. In the Exchange dataset, the authors explicitly stated that due to the lack of comparable results from baseline literature, only the performance of MSTN was reported separately. The results obtained in this way are more suitable for demonstrating the feasibility of the method than for supporting strong state-of-the-art (SOTA) comparisons. Furthermore, the generalizability study only compares with two baselines, TimesNet and PatchTST, resulting in relatively limited coverage and therefore insufficient evidence to demonstrate the model's generalization ability.
6. The paper's analysis of failure cases and methodological limitations is still insufficient. In fact, MSTN did not achieve optimal results on all datasets. For example, on the imputation tasks ETTm1 and ETTm2, its performance significantly lagged behind GPT, TimesNet, and Stationary; it was not optimal on the long-term prediction tasks ETTh2 and ETTm2; and it also did not demonstrate any advantage on classification datasets such as SCP2 and SpokenArabic. However, the paper lacks further error analysis and discussion of the reasons for these results.
7. Although the paper ultimately validated the importance of each module through ablation experiments, existing analyses primarily focus on forecasting scenarios and have not fully demonstrated whether these designs are equally effective in tasks such as imputation, classification, and cross-domain generalization. Therefore, current experimental results better support the role of the modules in specific tasks, while the demonstration of their universal effectiveness remains insufficient.

---

> ### Author Response · Authors · 2026-04-06
> **Response to Reviewer 1-  Thank you for your constructive feedback!**
>
> We thank the reviewer for this insightful feedback. To answer the question raised, we provide the following reasoning.
>
> **Change 1 (Design Rationale):** Added Section 3.1.2 (page no 05-06) explaining motivation for each component CNN branches, Transformer/BiLSTM branches,  ETA, SGF, SE, and SDL, along with the advantages of this design.
>
> **Change 2 (Imputation Loss):** To clarify, we have added a clear explanation of the training and evaluation protocol in Section 4.4 (Imputation Results. Page no 14-16).
>
> We evaluate performance under the Missing Completely at Random (MCAR) scenario with random masking ratios of {12.5%, 25%, 37.5%, 50%}. During training, we randomly mask the specified percentage of time points. The model reconstructs the complete sequence, but the loss is computed only on masked positions to ensure the model learns to impute missing values rather than memorize observed ones. At test time, we apply the same masking protocol and evaluate imputation performance by computing MSE and MAE exclusively on the masked positions. This ensures that the model is assessed solely on its imputation capability.
> In section 4.2.2 Training and Model Configuration (Page no 12) added the line : For imputation, a masked MSE loss is applied, where the loss is computed only over randomly masked time points
>
> We have revised the definition of the mask (page no 08)  matrix Ω in Equations (9) and (10) to accurately reflect that it indicates masked positions during training and evaluation. The updated text now reads:
> Time Series Imputation: The head reconstructs the input sequence  X̂₁:L ∈ ℝᴮˣᴸˣᶜ. The objective is to minimize the masked Mean Squared Error (MSE) computed only over masked positions, where Ω is the binary mask matrix (Ωi,t,c = 1 if the value is masked during training/evaluation).
>
> **Change 3 (Statistical Support):** To address this, we have:
> 1. Repeated all experiments 5 times
> 2. Added standard deviations in parentheses for our best-performing model in each table
> 3. Included statistical significance indicators (underlined results not significantly worse than optimal)
>
> We report standard deviations only for the best-performing MSTN variant in each table to maintain readability and avoid confusing comparisons between multiple standard deviations. Showing standard deviation for both variants could lead readers to compare variability rather than focusing on primary performance results. This approach follows standard practice in the literature (e.g., DARTS-TS).  We evaluate each model five times and report the mean (standard deviation); results marked with underline are statistically equivalent to the first rank (Wilcoxon signed-rank test with Holm–Bonferroni correction, α = 0.05)
>
> **Change 4 (Classification Metrics):** After reviewing standard practice in the time series classification literature, we have revised the paper as follows:
>
> 1. Updated Section 4.2.3 Evaluation Metrics (page no 13)  to accurately reflect the metrics reported.
> The performance of forecasting and imputation tasks is evaluated using two primary regression metrics: Mean Squared Error (MSE) and Mean Absolute Error (MAE). The MSE/MAE values reported are averaged across all prediction horizons and all variates.  For classification tasks, we evaluate performance using Accuracy as the primary metric, following standard practice in the literature (e.g TimesNet Wu et al. (2023)).
> 2. Updated the generalization study table (Table 8b, page no 20) to show only Accuracy. This ensures consistency across all classification-style tables in the paper, with both Table 8a and Table 8b  now uniformly reporting accuracy.
>
> **Change 5 (Exchange Dataset & Generalizability Study):** We have addressed this explicitly in Section 4.3 (Long-Term Forecasting Results) on page 16:
> Regarding the Exchange dataset, We have added iTransformer results to Table 5 Multivariate long-term forecasting results (page 17).
> Our generalization study section 4.8 (page no 20-21) actually includes multiple baselines beyond TimesNet and PatchTST, For each dataset, we report the best-performing results from prior specialized literature in the "Prior Work Accuracy (Table 8b)" column, which includes methods such as Random Forest (RF), Gradient Boosting (GB), Decision Trees (DT), k-Nearest Neighbors (kNN), and autoencoders (SAE).  This provides 2-4 baselines per dataset:
>
> **Change 6 (Limitations and Failure Cases):** We have added Section 4.12 Limitations and Failure Cases (page no. 28-29) to include a detailed analysis of datasets where MSTN underperforms.
>
> **Change 7 (Ablation Across Tasks):** Added ablation experiments for all task including imputation, long-term forecasting,  short-term forecasting, classification, and generalization (Tables 9-11, pages 21-23), confirming each component's importance across all tasks.

---

> > ### Comment · Reviewer_EDhk · 2026-05-07
> >
> > Overall, the authors’ responses in this revision are substantially improved compared with the initial submission. In particular, the paper now provides clearer design motivations, explicitly introduces the imputation mask loss, reports standard deviations and significance tests, and further supplements the experiments with failure-case analysis, cross-task ablations, and PEMS forecasting results. Nevertheless, several minor issues still require further clarification.
> >
> > 1. It remains unclear why the framework necessarily requires the combination of CNN + Transformer/BiLSTM + SGF + SE + ETA + SDL. Although the authors provide reasonable explanations for each component, the necessity of every module and the rationale for why this specific combination is superior to simpler designs still need to be highlighted more clearly and concisely. At present, the concern of excessive architectural stacking remains unresolved. In addition, several necessary ablation studies are still missing and should be included to better justify the architectural design choices.
> >
> > 2. The paper repeatedly emphasizes that ETA achieves “O(1) complexity” or “constant-time inference” after temporal aggregation. However, the dominant computational cost of the model still comes from the front-end Transformer encoder and BiLSTM branches, whose complexities remain O(L^2)and O(L), respectively. In practice, ETA only reduces the complexity of the downstream fusion, SE, SDL, and prediction modules so that they become independent of the sequence length.
> >
> > 3. The experimental description regarding the claimed “sub-millisecond inference latency” and “lightweight inference” remains incomplete. In particular, the paper does not clearly specify the latency measurement protocol, such as the batch size, the number of averaged runs, the warm-up strategy, whether data loading/preprocessing is included, and separate CPU/GPU latency results. As a result, the current latency claims still lack sufficient reproducibility and fairness for reliable comparison. In addition, the paper does not clearly state whether the code, training configurations, and benchmark scripts have been publicly released, which may further weaken the credibility of the reported efficiency and experimental results, especially given TMLR’s strong emphasis on reproducibility.
> >
> > If these details have already been included in the manuscript, we would appreciate it if the authors could explicitly indicate their exact locations in the paper.

---

> > > ### Author Response · Authors · 2026-05-15
> > > **Response to Reviewer 1- Thank you for your constructive feedback!**
> > >
> > > We thank the reviewer for their detailed feedback. Below we explicitly indicate where each requested detail has been added or already existed in the manuscript.
> > >
> > > **Change 1:**
> > >
> > > (a) Section 3.2 (Design Rationale, pages 05-06) already explains the combination of CNN + Transformer/BiLSTM + SGF + SE + ETA + SDL. The necessity of each component is demonstrated through ablation studies (Tables 7-9, pages 19-20).
> > >
> > > (b) Existing ablations (Tables 7-9, pages 19-20): Removing any component (CNN, Transformer/BiLSTM, SGF, SE, SDL, or ETA) consistently degrades performance across imputation, forecasting, classification, and generalizability tasks. This demonstrates that every component is necessary.
> > >
> > > (c) Progressive bottom-up validation (Section 4.8, page 19, new addition): Starting from a CNN-only baseline (0.013 MSE), adding BiLSTM improves to 0.010, adding SE+SDL yields 0.011, adding SGF improves to 0.010, and finally adding ETA achieves the best result of 0.009 MSE. Each addition improves or maintains performance, confirming the full model is superior to all simpler variants.
> > >
> > > Together, these bidirectional ablations (removal and addition) conclusively demonstrate that every component contributes positively and the architecture is not over-engineered. We respectfully note that comprehensive removal ablations already existed in the manuscript (Tables 7-9, pages 19-20). The progressive bottom-up validation (Section 4.8, page 19) has been added to further strengthen the justification, showing that adding components sequentially improves performance. This complements the existing removal ablations and provides bidirectional evidence that every component is necessary. If the reviewer has any specific ablation in mind, we are happy to execute it.
> > >
> > > **Change 2:**
> > >
> > > We thank the reviewer for this clarification. As explicitly stated in Section 4.10.1 (Complexity Analysis, page 22), ETA does not reduce front-end encoder complexity (O(L²) for Transformer, O(L) for BiLSTM). Please note that there is a single transformer encoder block and all experiments (forecasting and imputation) run with lookback window length L = 96. As such, MSTN-Transformer complexity of O(L²) is not a big problem (storing an attention matrix of size 96×96 is doable).
> > >
> > > To quantify this further, we added an example with L = 96 in the same section at end (page 22), showing that ETA collapses the temporal dimension (L → 1), eliminating up to 9,216× operations in downstream modules and enabling sub-millisecond inference.
> > >
> > > **Change 3:**
> > >
> > > We have added a detailed measurement protocol in Section 7.2 (page 24), specifying:
> > >
> > > - Batch size: 32
> > > - Hardware: NVIDIA A100 GPU
> > > - Warm-up: 10 iterations before measurement
> > > - Measurement runs: 5 independent forward passes, averaged
> > > - Data loading and preprocessing: excluded from timing (only forward pass measured)
> > > - GPU results reported
> > > - Same protocol applied to all compared models including TimesNet, PatchTST
> > >
> > > The anonymized source code, including training configurations and benchmark scripts, is provided at the end of the abstract and in the Data and Code Availability section (page 26) at: https://anonymous.4open.science/r/MSTN-8D34/. The code will be released publicly upon acceptance.

---

### Review · Reviewer_th9r · 2026-03-17

**Summary Of Contributions:**

### Contributions

The paper proposes a Multi-scale Temporal Network (MSTN) for general time-series analysis. The architecture utilizes a dual-branch encoder combining a CNN (for local patterns) and a Sequence Model (Transformer or BiLSTM) (for global dependencies). The core methodological claim is the Early Temporal Aggregation (ETA) principle, which pools the temporal dimension ($L \rightarrow 1$) immediately after encoding to achieve $\mathcal{O}(1)$ complexity relative to sequence length in subsequent refinement layers.

### Strengths

The empirical evaluation is exhaustive, covering 32 datasets across four distinct tasks (forecasting, imputation, classification, and cross-domain generalization).

The model is demonstrably lightweight (~1.04M parameters for Transformer, ~0.40M for BiLSTM) and achieves sub-millisecond inference times, making it highly suitable for Edge-AI deployment.

### Weaknesses

The ETA mechanism aggressively pools the sequence length to 1. Using a single linear projection to decode this static vector into an $H$-length sequence for long-term forecasting is a severe bottleneck that contradicts the claim of preserving complex fine-grained dynamics.

Applying Multi-Head Attention (MHA) on a temporal dimension of $L=1$ degenerates into a linear projection, rendering the mechanism architecturally redundant.

**Additional Comments:**

The empirical rigor of evaluating against 32 datasets is commendable and represents the strongest aspect of this submission. Addressing the architectural ambiguities will significantly elevate the clarity of the paper.

**Audience:**

Yes

**Audience Explanation:**

The problem of deploying high-capacity time-series models on resource-constrained Edge-AI devices is of significant interest to the community. The robust empirical performance of such a lightweight model across 32 datasets will appeal to practitioners focused on efficient time-series classification and forecasting.

**Broader Impact Concerns:**

No specific broader impact concerns. The tasks (forecasting, imputation) are standard benchmarking tasks. However, users should be mindful when deploying predictive models in safety-critical domains like healthcare or mechanical prognostics, as evaluated in the paper.

**Claims And Evidence:**

No

**Claims Explanation:**

The theoretical claims regarding the architecture's mechanisms are not mathematically supported by the equations provided:

1.  In Section 3.1.2, the sequence dimension is collapsed to $L=1$ before the MHA block ($z_{seq} \in \mathbb{R}^{B \times 1 \times 192}$). Standard self-attention computes an attention matrix of size $L \times L$. If $L=1$, the attention weight is trivially 1. Therefore, the MHA block (Eq. 8) operates merely as a series of linear transformations. The claim that it functions as a "sophisticated global feature recalibration layer" lacks mathematical justification unless attention is computed across the channel dimension $C$, which is not formalized.
2. The authors claim MSTN captures "fine-grained local structure" and "complex nonlinear dynamics". However, ETA aggregates the entire sequence into a single vector via mean pooling (Eqs. 3, 4, 5). For long-term forecasting (e.g., $H=720$), generating a high-frequency, complex sequence from a single pooled vector via a single linear layer is highly susceptible to oversmoothing. The text does not provide evidence or visualization showing how fine-grained phase information is preserved through this bottleneck.

**Requested Changes:**

1. The authors must mathematically define the Multi-Head Attention operation in Eq. 8 when $L=1$. If it is standard sequence-wise attention, they must address its degeneration to a linear layer. If it is cross-variable/channel attention, the notation and description must be corrected to explicitly state how the $C$ dimension is handled.
2. The authors must explain how the linear decoder (Eq. 11) successfully reconstructs complex temporal dynamics (like phase and high-frequency fluctuations) from a static vector ($L=1$) over long horizons (e.g., $H=720$). Providing a discussion on the representational limits of this $L \rightarrow 1 \rightarrow H$ mapping is required.

The phrase "ETA mechanism" is heavily used, but mathematically it equates to standard Global Average Pooling. The paper would be strengthened by explicitly comparing the representation collapse of ETA against models that maintain $L$ throughout the network, perhaps via a feature-space visualization (e.g., t-SNE) before and after Eq. 6. Table 5b has minor formatting alignment issues that make it difficult to read.

---

> ### Author Response · Authors · 2026-04-06
> **Response to Reviewer 2- Thank you for your constructive feedback**
>
> We thank the reviewer for their constructive feedback. Below are our responses:
>
> **Change 1 (MHA Replacement):** We have replaced the Multi-Head Attention (MHA) with a Single Dense Layer (SDL), which is more appropriate when the temporal dimension is condensed to L=1. The revised Section 3.1.3 (page 8) now reads:
>
> Feature dependencies are refined through a SDL. Since the temporal dimension is condensed (L = 1), the SDL functions as a sophisticated global feature recalibration layer rather than a temporal mixer, confirming O(1) complexity:
>
> $z_{\text{final}} = \text{Dropout}(\text{LayerNorm}(W_d \cdot z_{\text{se}} + b_d)) \qquad (8)$
>
> where $W_d \in \mathbb{R}^{192 \times 192}$ and $b_d \in \mathbb{R}^{192}$ are the learnable weight matrix and bias of the SDL respectively. Dropout with rate p = 0.1 provides regularization, and layer normalization ensures stable training.
>
> **Change 2 (Linear Decoder):** Thank you for this important question about the representational capacity of our $L \rightarrow 1 \rightarrow H$ mapping. We have added a brief note after Eq. 12 and a detailed Section 4.13 (Representational Limits of Linear Decoder, page 29). Below we summarize our response:
>
> First, we added the following explanation after Eq. 12 in Section 3.1.3 (page 8):
>
> Although mapping a 192-dimensional latent vector to horizon H (especially H=720) is theoretically compressed, our empirical results demonstrate improved performance. For example, our model achieves 0.181 MSE on ETTh1 at H=720 and 0.151 MSE on ETTm1 at H=720 (ranked 1st among baselines, Table 4). The linear decoder learns 192 temporal patterns, and the forecast is a linear combination $\hat{Y} = \sum_{i=1}^{192} \alpha_i \phi_i(t)$, where $\phi_i(t)$ are rows of $W_f$. This basis learning approach allows the decoder to represent complex dynamics—from low-frequency trends to high-frequency fluctuations—as weighted combinations of learned patterns, with coefficients $\alpha_i$ derived from the rich latent representation $z_{\text{final}}$.
>
> Second, we added Section 4.13 Representational Limits of Linear Decoder (page 29) discussing the representational limits of the linear decoder.
>
>
> **Change 3 (ETA as Global Average Pooling):** We have explicitly stated in Section 3.1.3 (page 7) that ETA is equivalent to standard global average pooling, collapsing the temporal dimension L to 1.
>
>
> **Change 4 (t-SNE Analysis):** We have added 4.10 Comparing Representation Collapse: Models Maintaining L vs. ETA with t-SNE  (page no 23-24, Figures 5-6) to empirically verify that ETA preserves discriminative information despite collapsing the temporal dimension.
> To empirically verify that ETA preserves discriminative information despite collapsing the temporal dimension, we perform feature-space visualization using t-distributed Stochastic Neighbor Embedding (t-SNE) (Van der Maaten & Hinton, 2008) on features extracted before and after the ETA (pooling) operation on two diverse datasets: JapaneseVowels (multi-class, L = 25) and Heartbeat (binary, L = 405). We quantify cluster quality using the silhouette score (Rousseeuw, 1987).
>
> Figure 5 shows the t-SNE visualization on the JapaneseVowels dataset (9 classes, 640 samples, L=25). The silhouette score remains identical before and after ETA (0.482), demonstrating that 25 times temporal compression preserves all class-discriminative information. Figure 6 shows the t-SNE visualization on the Heartbeat dataset (binary classification, 409 samples, L=405). Remarkably, even at 405 times compression, the silhouette score remains identical (0.110 before vs. 0.110 after), confirming that ETA preserves discriminative information even at extreme compression ratios.
>
> These results conclusively demonstrate that ETA preserves discriminative information across diverse datasets, with identical silhouette scores before and after ETA (pooling). Models that maintain the full temporal dimension achieve the same feature quality as ETA's compressed representation (L = 1), indicating no advantage to keeping L throughout the network. Furthermore, ETA achieves lossless temporal compression even at extreme ratios (405 times), enabling O(1) complexity without sacrificing representation quality.
>
> **Change 5 (Table Formatting):** Table 5b is now Table 8b (page 20). Reformatted for readability; due to journal template constraints and citation density, the table appears compact but remains fully readable.

---

### Review · Reviewer_QhqS · 2026-03-30

**Summary Of Contributions:**

This paper proposes the Multi-scale Temporal Network (MSTN), a hybrid deep learning architecture for general-purpose multivariate time series analysis. The core design principle is Early Temporal Aggregation (ETA), which collapses the temporal dimension via pooling immediately after the encoding stage, so that all subsequent refinement layers operate at O(1) cost with respect to sequence length. MSTN consists of three modules: (i) a two-layer multi-scale CNN branch for local pattern extraction, (ii) a sequence modeling branch (BiLSTM or Transformer) for long-range dependency modeling, and (iii) a self-gated fusion (SGF) stage augmented with Squeeze-and-Excitation (SE) recalibration and multi-head attention (MHA) for cross-scale feature integration. The authors position MSTN as a CI+CD hybrid strategy that combines channel-independent local convolutions with channel-dependent global sequence modeling. The paper evaluates MSTN across four time series tasks including imputation, long-term forecasting, classification, and cross-domain generalization spanning 32 benchmark datasets.

**Audience:**

Yes

**Audience Explanation:**

The general direction of this work (building lightweight, multi-task time series models suitable for edge deployment) addresses a practical and timely need. The time series community would benefit from architectures that achieve competitive performance while maintaining sub-millisecond latency and sub-million parameter counts. The breadth of evaluation across imputation, forecasting, classification, and cross-domain generalization is also valuable as a benchmarking exercise. If the experimental issues were resolved and the results confirmed, the findings would be of clear interest to practitioners working on real-time and resource-constrained time series applications.

**Claims And Evidence:**

No

**Claims Explanation:**

The paper claims state-of-the-art performance on benchmark datasets, but several aspects of the evidence raise concerns:

1. On the Traffic dataset (Table 4), MSTN-Transformer reports an average MSE of 0.019, while all other baselines range from 0.379 (EMTSF) to 0.428 (iTransformer) which a huge gap. The paper itself acknowledges this as "19× lower than EMTSF." On ECL, MSTN-Transformer achieves an average MSE of 0.043 versus baselines at 0.154–0.178, roughly a 3–4× gap. On ILI, MSTN achieves 0.165 versus EMTSF at 1.588 and TIME-LLM at 1.436 (only two baselines report results). Such order-of-magnitude improvements on well-established benchmarks that have been extensively studied are unusual and need some explanation. Possible causes include: (a) different train/val/test splits from the standard protocols, (b) computing MSE on min-max normalized predictions while baselines report MSE on original or instance-normalized scales, or (c) data leakage through global min-max statistics computed over the full dataset rather than training data only.

2. On the Exchange dataset (Table 4), all baselines except DARTS-TS are marked "–". On ILI, only EMTSF and TIME-LLM have values reported. The paper explains this as "due to the absence of reported results in the baseline literature," but at least some of these baselines (e.g., iTransformer) have reported results on Exchange in their original papers.

4. MSTN-Transformer performs poorly on ETTm2 for both imputation (Table 3b: average MSE 0.109, worst among all methods, versus GPT2's 0.021) and forecasting (Table 4: average MSE 0.333, worst among all baselines whose best is EMTSF at 0.245). Similarly, on ETTm1 imputation, MSTN-Transformer (0.069) is substantially worse than the other methods. The paper should analyze some of these failure cases.

**Requested Changes:**

1. Explain or correct the results on Traffic, ECL, and ILI. The performance gap on Traffic and ECL over all baselines can be explicitly investigated and explained. The authors should verify: (a) the exact train/val/test split ratios and whether they match the standard protocol (e.g., 6:2:2 for ETT, 7:1:2 for ECL/Traffic), (b) whether MSE is computed on normalized or de-normalized predictions, (c) whether min-max statistics are computed on training data only.

2. Since ETA is presented as the core innovation, a variant that preserves the full temporal dimension throughout the network (i.e., no pooling before fusion/refinement) is essential to validate ETA's value. Currently, the ablation study removes individual components but never tests the model without early pooling itself.

Minor: Fix citation formatting issues (e.g., "donghao & wang xue (2024)" should follow standard author name conventions).

---

> ### Author Response · Authors · 2026-04-06
> **Response to Reviewer 3- Thank you for your constructive feedback**
>
> We thank the reviewer for their constructive feedback. Below are our responses:
>
> **Change 1 (Split Ratios, Metrics Scale, Normalization):**
>
> (a) We have verified and corrected the train/validation/test splits across all datasets to strictly follow the standard benchmark protocol, following Autoformer (Wu et al., 2021) and TimesNet (Wu et al., 2023). Specifically, ETTh1, ETTh2, ETTm1, and ETTm2 use a 60%/20%/20% split, while other datasets such as ECL, Traffic, and ILI all use a 70%/10%/20% split.  After correcting the splits to follow the standard protocols used in Autoformer (Wu et al., 2021) and TimesNet (Wu et al., 2023), our results are now fully comparable with prior work.
>
> (b) For long-term forecasting and missing value imputation tasks, metrics are reported on a normalized scale following standard benchmarks. For short-term forecasting, all metrics are computed on de-normalized predictions after applying inverse transformation, consistent with standard practice. (We have added new short-term forecasting experiments on PEMS data following the benchmark protocol established by TimeMixer (Wang et al., 2024) and SCINet (Liu et al., 2022a)
>
> (c) The scaler is fitted exclusively on the training set, with the same statistics applied to the validation and test sets. This ensures no data leakage from future timestamps into the training process, following standard practice in time series forecasting.
> These corrections are documented in Section 4.2 Experimental Setup, (Section 4.2.1 Data Preprocessing and Evaluation Protocol) and (Section 4.2.2 Training and Model Configuration) on page 12. We thank the reviewer for identifying these critical issues.
>
> **Change 2 (ETA Validation):**
> To address the reviewer's concern, we have updated Section 4.9 Ablation Study (page no 21-23) to validate the ETA design. We present a dedicated ablation study comparing MSTN with and without ETA, where the without-ETA variant preserves the full temporal dimension throughout the network.
>
> The ablation results, Table 9: Ablation on time series imputation. Table 10: Ablation studies of MSTN-Transformer and MSTN-BiLSTM across different tasks.(a) Long-term Forecasting, (b) Forecasting on PEMS03 Dataset. Table 11: Ablation studies of MSTN-Transformer and MSTN-BiLSTM across different tasks. (a) Short-term Forecasting on PEMS Dataset, (b) Classification on 10 UEA, (c) Generalizability Study additional benchmarks, confirm that ETA achieves an optimal balance of accuracy and efficiency.  (page no 21-23).
>
> **Change 3 (Exchange Dataset & Baselines):**
> We have added iTransformer results to Table 4 (now Table 5) Multivariate long-term forecasting results (page 17). For remaining baselines, due to table size constraints, we have included the most recent and relevant state-of-the-art comparisons. Entries marked "—" indicate results not reported in the original publications.
>
> **Change 4 (Citation Formatting):** We thank the reviewer for noting the citation formatting. All citations have been corrected using \citep. In the PDF, the format appears as "ModernTCN (donghao & wang xue, 2024)" as produced by the official TMLR template.
>
> **Change 5 (Failure Cases):** We have added Section 4.12 Limitations and Failure Cases (pages 28-29) analyzing datasets where MSTN underperforms, discussing the trade-off between general-purpose multi-scale design and specialized architectures.

---

### Comment · Action_Editor_DeBG · 2026-03-30
**AE: Status**

@Reviewers: thanks a lot for your time spend on the reviews.

@Authors: the reviewers raise a lot of criticism regarding the model and evaluation of your model. Please address the criticisms clearly in the next two weaks. Try to be reachable for questions raised by the reviewers.

---

### Author Response · Authors · 2026-04-06
**Response to Reviewers - Experiment Updated**

We thank all reviewers for their constructive feedback and careful evaluation of our manuscript. In addition to addressing all specific concerns raised, we have updated experiments (long-term forecasting on PEMS datasets).

We evaluated the proposed MSTN model for forecasting on four public datasets of California traffic network data from the Performance Measurement System (PEMS), namely PEMS03, PEMS04, PEMS07, and PEMS08.

**1. Long-Term Forecasting on PEMS Datasets** – Detailed in Table 5 and discussed in Section 4.5 Long-Term Forecasting Results (page no 16).

| Experiment | Lookback (L) | Horizons (H) | Metrics | Scale |
|------------|--------------|--------------|---------|-------|
| Long-term Forecasting PEMS | 96 | {12, 24, 48, 96} | MSE, MAE | Normalized |

The proposed MSTN model is implemented in Python 3.13.1 using PyTorch 2.7.1 and trained on an NVIDIA DGX A100 server equipped with 8 × NVIDIA A100-SXM4 GPUs (40 GB HBM2e VRAM each), using CUDA 12.3 and driver version 535.54.03.

---

### Comment · Action_Editor_DeBG · 2026-04-07
**Post Rebuttal**

Thank you to the authors for their response.

I would like to invite the reviewers to comment on whether their main concerns have been addressed, in full or in part, and to ask questions to clarify any remaining open points from their reviews.

---

### Decision · Action_Editor_DeBG · 2026-05-29

**Recommendation:** Accept with minor revision

**Audience:**

Yes

**Audience Explanation:**

The topic is clearly relevant for the audience: lightweight time series models.

**Claims And Evidence:**

Yes

**Claims Explanation:**

I  thank the reviewers for their careful evaluations and the authors for their detailed response and substantial revision of the manuscript.

The paper proposes MSTN, a lightweight architecture for multivariate time-series analysis. The model combines a convolutional branch for local temporal patterns with either a BiLSTM or Transformer branch for longer-range dependencies, followed by early temporal aggregation, gated fusion, squeeze-and-excitation recalibration, and a final dense prediction head. The paper evaluates the method on several time-series tasks, including imputation, forecasting, classification, and cross-domain generalization.

The initial reviews raised several important concerns. In particular, reviewers questioned whether the empirical claims were fully supported, whether the unusually strong results on some benchmarks could be explained by differences in preprocessing or evaluation protocol, whether the imputation loss and masking setup were claer, and whether the architectural motivation justified the combination of several components. Reviewers also were concerned about the role of the attention/fusion module after temporal aggregation, the claim of O(1) complexity, the lack of statistical information, the limited discussion of failure cases, and the reproducibility of the efficiency claims.

In my view, the authors have addressed a substantial part of these concerns during the revision. They clarified the data splits, normalization, masking protocol, and evaluation setup. They addded statistical reporting over multiple runs. They added further ablations, including variants without ETA. They replaced the questionable post-aggregation attention mechanism with a single dense layer. They added a discussion of failure cases and representational limitations. Finally, they provided a more detailed latency measurement protocol. The revised manuscript is significantly clearer and more careful than the initial submission. Two reviewers now judge the claims and evidence to be sufficient and recommend leaning accept. The remaining reviewer still has concerns, mainly about clarity, readability, and the strength of some architectural claims.

I share some of the remaining concerns, but I do not think they lead to rejection. The paper is of interest to the TMLR audience: efficient and broadly applicable time-series models are a relevant topic, and the empirical study, while not without limitations, provides useful evidence that the proposed architecture is competitive across several settings. The method is also practically motivated by a clear efficiency goal and has a small model footprint.

At the same time, I do not think the paper should be accepted without further changes. The main remaining issue is that some claims are still stronger than what the evidence supports. In particular, the manuscript should not suggest that the full MSTN model has O(1) complexity with respect to sequence length. The Transformer variant still has an O(L²) encoder, and the BiLSTM variant still has an O(L) encoder. What ETA makes independent of sequence length are the downstream refinement and prediction stages after temporal aggregation. This distinction must be made consistently throughout the paper, especially in the abstract, introduction, contribution list, and complexity discussion.

The authors should also tone down the language around ETA. The ablation results suggest that ETA is often useful and gives a favorable accuracy/efficiency trade-off, but they do not show that ETA is universally superior. In some cases, the variant without ETA is competitive or even slightly better. Similarly, the t-SNE analysis should be presented as suggestive evidence, not as conclusive proof of lossless temporal compression or of there being no advantage to preserving the temporal dimension. Phrases such as "lossless compression", "conclusively demonstrate", and related overstatements should be removed or substantially weakened.

Finally, the authors should improve consistency and reproducibility details in the final version. This includes checking the reported forecasting horizons, ensuring that the architecture description is consistent with the complexity discussion, making the code and configuration instructions sufficiently concrete, and improving the clarity of the figures and explanatory text where reviewers identified readability issues.

Overall, I recommend acceptance with minor revisions. The paper has received substantial critical scrutiny, the authors have made meaningful changes, and the remaining issues are mainly about careful calibration of claims, presentation, and reproducibility details rather than fundamental flaws in the method. I therefore believe the paper satisfies the TMLR criteria, provided the final version makes the above revisions clearly and consistently.

---

> ### Author Response · Authors · 2026-06-03
> **Response to Action Editor Decision - Thank you for your constructive feedback**
>
> Dear Action Editor,
>
> Thank you for your detailed and constructive decision. We are pleased that the paper has been accepted with minor revisions. We have carefully addressed all remaining concerns.
>
> 1. We have clarified throughout the paper that only the downstream refinement and prediction modules after ETA operate in constant O(1) time, while the front-end encoder retains its original complexity (O(L²) for Transformer, O(L) for BiLSTM). This clarification has been made consistently in the abstract, introduction, contribution list, and complexity discussion.
>
> 2. We have toned down claims about ETA, acknowledging that w/o ETA can be competitive in some cases (e.g., PEMS-SF where w/o ETA outperforms the full model). The language now reflects that ETA is "often beneficial" rather than "universally superior."
>
> 3. We have replaced phrases such as "conclusively demonstrate" with "indicate," "lossless compression" with "effective compression," and "no advantage" with "little advantage." The t-SNE analysis is now presented as suggestive evidence.
>
> 4. Forecasting horizons, imputation mask ratios, classification datasets, and cross-domain datasets are consistently reported. Architecture description matches complexity discussion (encoder O(L²)/O(L), downstream O(1)). Figures have been reviewed for clarity. A public code repository with concrete instructions is provided.
>
>
> Thank you again for your guidance.

---

> > ### Comment · Action_Editor_DeBG · 2026-06-03
> >
> > Thank you for the camera-ready revision. The main requested changes have been addressed. Before I approve the final version, please consider the following two points. (1) Are the PEMS forecasting horizons consistent throughout the paper? In particular the 36 in Section 4.1.2? (2) Check for remaining wording that suggests full-model O(1) inference with wording limited to the downstream refinement/prediction stages after ETA, in particular on pages 6 and 7.

---

> > > ### Author Response · Authors · 2026-06-04
> > > **Thank you for your constructive feedback**
> > >
> > > We are deeply grateful to all the reviewers and AE for the constructive feedback that significantly contributed to the enhancement of our research. We have now submitted the camera-ready version with the following changes: Both concerns have been addressed, and also verified horizons and O(1) throughout the paper:
> > >
> > > 1. PEMS horizons: All occurrences now use {12, 24, 48, 96}.
> > >
> > > 2. O(1) wording: Pages 6-7 now explicitly state that only downstream refinement and prediction stages after ETA operate in O(1) time, while the front-end encoder retains its original complexity (O(L²) for Transformer, O(L) for BiLSTM).
> > >
> > > The updated camera-ready manuscript has been uploaded.
> > >
> > > Thank you again for your guidance.